# *Drosophila melanogaster* Chemosensory Pathways as Potential Targets to Curb the Insect Menace

**DOI:** 10.3390/insects13020142

**Published:** 2022-01-28

**Authors:** Md Zeeshan Ali, Anwar L. Bilgrami, Jawaid Ahsan

**Affiliations:** 1Drosophila Behavioral Laboratory, Department of Biotechnology, Central University of South Bihar, Gaya 824236, India; zeeshanali@cusb.ac.in (M.Z.A.); anushreebtn@cusb.ac.in (A.); 2Deanship of Scientific Research, King Abdulaziz University, Jeddah 21589, Saudi Arabia; alegman@kau.edu.sa

**Keywords:** *Drosophila melanogaster*, gustation, gustatory receptors, insect repellents, insect vectors, odorant receptors, olfaction, signaling pathway

## Abstract

**Simple Summary:**

The perception and processing of chemosensory stimuli are indispensable to the survival of living organisms. In insects, olfaction and gustation play a critical role in seeking food, finding mates and avoiding signs of danger. This review aims to present updated information about olfactory and gustatory signaling in the fruit fly *Drosophila melanogaster*. We have described the mechanisms involved in olfactory and gustatory perceptions at the molecular level, the receptors along with the allied molecules involved, and their signaling pathways in the fruit fly. Due to the magnifying problems of disease-causing insect vectors and crop pests, the applications of chemosensory signaling in controlling pests and insect vectors are also discussed.

**Abstract:**

From a unicellular bacterium to a more complex human, smell and taste form an integral part of the basic sensory system. In fruit flies *Drosophila melanogaster*, the behavioral responses to odorants and tastants are simple, though quite sensitive, and robust. They explain the organization and elementary functioning of the chemosensory system. Molecular and functional analyses of the receptors and other critical molecules involved in olfaction and gustation are not yet completely understood. Hence, a better understanding of chemosensory cue-dependent fruit flies, playing a major role in deciphering the host-seeking behavior of pathogen transmitting insect vectors (mosquitoes, sandflies, ticks) and crop pests (*Drosophila suzukii*, Queensland fruit fly), is needed. Using *D. melanogaster* as a model organism, the knowledge gained may be implemented to design new means of controlling insects as well as in analyzing current batches of insect and pest repellents. In this review, the complete mechanisms of olfactory and gustatory perception, along with their implementation in controlling the global threat of disease-transmitting insect vectors and crop-damaging pests, are explained in fruit flies.

## 1. Introduction

*Drosophila melanogaster* has provided a new dimension to the discipline of neuroscience. The research on olfactory and gustatory systems using fruit flies has grown rapidly during the last few years due to the strong genetic similarity that humans share with them. This has proved instrumental in establishing *D. melanogaster* as an ideal model organism. Additionally, fruit flies offer additional advantages to researchers over humans or mammals. The flies can be easily cultured in laboratory conditions and are quite inexpensive to maintain. They produce a large number of eggs and have a relatively short life cycle that provides much-needed flexibility to the researchers. The flies can also be genetically modified [1] and the neurons easily detected in every individual fly [2]. The larval stages of the fly can be used for different experiments, since they are simple in structural organization and easy to handle as compared to adult flies.

Olfaction plays a vital role in the survival of *D. melanogaster*. The process commences with the binding of volatile odorant molecules to their specific receptors known as odorant receptors (Ors), which are expressed on the olfactory sensory neurons (OSNs). In the larval stages, each of the two dorsal organs (DO) have 21 OSNs [3,4], which contrarily, are contained in specialized sensory hairs known as sensilla, situated on the third antennal segments and maxillary palps in the adult flies. Earlier studies on olfaction in *D. melanogaster* used odorants in association with an appetitive or an aversive stimulus. The naïve *D. melanogaster* larvae showed vigorous chemotaxis towards many odorants, including ethyl acetate (EA) [5]. It was observed that the behavioral responses of larvae were diametrically opposite to shorter and longer chain acetates. The former (methyl to pentyl acetates) were attractive in nature, whereas the latter (hexyl to octyl acetates) triggered repulsion [6]. Besides this, the flies effectively avoided odorants which were paired with an electric shock in learning and memory experiments [7]. An age-dependent decline in olfactory response/memory was reported in *D. melanogaster* [8] and in those offspring born to aging flies [9]. These behavioral assays based on odorants have played a major role in discovering specific genes involved in the olfactory pathways [10,11] and the regions of the insect brain controlling them.

In gustation, there is a remarkable similarity of food choices and their detection that fruit flies share with mammals. This has given opportunities to researchers to conduct in-depth studies of taste perception. Both mammals and fruit flies consume carbohydrates as their major food source, and both avoid chemicals that are either toxic or taste bitter. The sensitivity and range of recognition of gustatory receptors (Grs) of *D. melanogaster* are similar to mammals. The Grs are mainly localized in the head and pharyngeal regions of the larvae [12], while in adult flies, they are dispersed on the mouthparts, leg tarsi, and around the female ovipositor. The wings of *D. melanogaster* also possess taste sensilla. The margins of the anterior wings respond to both appetitive and aversive stimuli due to increases in Ca^2+^ ions in the cytoplasm [13]. The Grs detecting food items are well characterized in *D. melanogaster* with a single Gr responding to fructose [14], but the case is different in bitter taste receptors. Several researchers have proposed bitter taste Grs to exhibit a multimeric organization comprising of one subunit tuned to a limited specificity, while others function as broadly required co-receptors [15,16,17,18,19,20]. The role of the intestinal gut of fruit flies in perceiving gustatory stimuli has also been reviewed. Due to the presence of Gr transcripts, the intestinal gut controls many biological functions such as ingestion, absorption of nutrients, and balancing body sugar levels [21].

The in-depth understanding of the chemosensory machinery of *D. melanogaster* has great practical applications, which may be implemented to design new methods of controlling disease-causing insect vectors and crop pests, and to gain a better understanding of how the batches of insect and pest repellents that are currently available in the market function. Insects are one of the major sources of the transmission of pathogens to humans and cattle, as well as the destruction of food crops [22]. Mosquitoes, ticks, sandflies, and mites are a few of the major insect vectors that spread life-threatening diseases to human beings, such as; malaria, dengue, West Nile fever, encephalitis, yellow fever, and Congo hemorrhagic disease [23]. In particular, mosquitoes are the biggest menace to the human population. *Anopheles* mosquitoes, i.e., *An. Gambiae* and *An. Funestus*, are involved in the transmission of the malarial parasite, *Plasmodium* sp., to their human hosts, triggering one million annual deaths [24,25]. The human filarial nematode *Wuchereria bancrofti*, and arboviruses spread through the bites of *Culex* mosquitoes [25] and *Aedes aegypti* (the yellow fever mosquito), account for many cases of dengue globally [25,26]. As regards crop pests, the facts are disturbing; for example, the spotted wing *Drosophila suzukii,* an invasive global insect pest, causes huge crop loss by destroying fresh, ripened small fruits and tree fruits at large scale [27,28,29,30].

To present remedies for such problems, the role of chemosensory mechanisms was evaluated in pest management using *D. melanogaster* as a model organism.

## 2. Olfactory System—Components and Basic Organization

In *D. melanogaster* larvae, the olfactory system is mainly localized to the head region. The head portion of larvae has a couple of DOs expressing 21 OSNs in each [3,4] (Figure 1A). Contrarily, the adult fruit flies detect odorants through a pair of antennae and maxillary palps (Figure 1B). These appendages are positioned on the head region, enveloped with numerous sensory hairs called sensilla. The sensilla possess OSNs that are specialized in detecting odorants (Figure 2A). Each antenna has about 410 olfactory sensilla with nearly 1300 OSNs, whereas the number of olfactory sensilla is up to 60 in each maxillary palp with approximately 120 OSNs [31,32,33,34]. These sensory hairs exhibit differences in their morphology. A recent work on serial block-face scanning electron microscopy (SBEM) images of antennal tissues has further led to a systematic morphological and morphometric analysis of the identified olfactory sensilla in *D. melanogaster*. A plethora of new information, such as inner dendritic enlargement with enhanced mitochondrial content, the presence of extracellular vacuoles in the lumen of sensilla, empty sensilla with no OSNs, and two new unconventional types of basiconic sensory hairs, was discovered. The olfactory sensilla in fruit flies can be separated into four different groups [35] (Figure 2B), which are as follows:

**a. Basiconic sensilla**: The sensory hairs at the proximomedial region of the antenna are club-shaped and are called basiconic sensilla. These sensilla are of 10 types, viz. ab1-ab10, are further grouped into three sub-classes, i.e., large, small, and thin. The large basiconic sensilla (ab1–ab3), ~12 µm in length, house two or four OSNs, whereas small basiconic sensilla (ab7–ab10), which are ~9 µm long, enclose two OSNs. The thin basiconic sensilla (ab4–ab6) are also ~12 µm in length but are thinner in shape. They encompass mostly two with a few cases of four OSNs per sensillum [31,41]. Additionally, there are also two novel types of basiconic sensilla; one is a large basiconic sensillum, abx(3), which encloses three OSNs, and the other is a small basiconic sensillum, abx(1), with a single OSN. Interestingly, one small basiconic sensillum with no OSN is also known to exist, which is designated as abx(0) [35]. The sensory hairs on the maxillary palp are all thin basiconic (pb1–pb3), each housing two OSNs [41]. The basiconic sensilla are single-walled.

**b. Trichoid sensilla**: The trichoid sensilla are pointed in shape and vary in length from 18–22 µm [31]. They occupy the lateral profile of the antenna, predominantly at the distal tip. These sensilla are of four types, namely at1-at4, and are classified as T1 (at1), T2 (at2), and T3 (at3 & at4) corresponding to the number of OSNs they house, i.e., one, two, and three, respectively [31,41]. However, Fluorescence-guided Single Sensillum Recording (FgSSR) analyses have re-categorized two trichoid sensilla, namely at2 and at3, as intermediate sensilla ai2 and ai3 [42]. Besides this, one trichoid sensillum belonging to the T1 subtype with no OSN has also been found in fruit flies and named T(0) [35]. The trichoid sensilla are also single-walled.

**c. Intermediate sensilla**: There are a few sensory hairs in the antenna exhibiting length and structure in between trichoid and basiconic sensilla. These are called intermediate sensilla. They are scattered among the trichoids on the frontal antennal surface, with their number varying from 10–20 on each antenna [31,41]. The intermediate sensilla are of two types, namely ai2 and ai3, housing two and three OSNs respectively [42]. They are single-walled.

**d. Coeloconic sensilla**: The coeloconic sensilla are the smallest of all the sensory hairs. They are short in size (~5 µm) and peg-shaped. Although they are dispersed with other groups of sensilla, the majority of them cover the posterior surface of the antenna. The number of OSNs in coeloconic sensilla varies between two to four neurons per coeloconic sensillum [31,41,43]. The ac3 sensillum houses two OSNs, ac2 and ac4, housing three OSNs each, whereas ac1 compartmentalizes four OSNs [35]. These sensory hairs are double-walled.

Further, in *D. melanogaster*, the antennal lobe (AL) functions as a deutocerebral neuropil showing functional similarities to the olfactory bulb of the vertebrate brain [44]. At larval stage, the antennal nerve (AN) connects 21 OSNs in DO to the larval antennal lobe (LAL). The LAL has about 30 subunits comparable to the glomerulus of an adult fly [4,45]. The sensory information from LAL then goes through several projections and local neurons (PNs & LNs) to be relayed to higher centers of the larval brain to initiate a behavioral response (Figure 3A). Similarly, in adult flies, OSNs located in the antenna relay sensory information to the AL, consisting of tightly packed neuropils called glomeruli. The number of glomeruli in *D. melanogaster* is 54 per AL, out of which 52 are innervated by chemosensory and 2 by thermosensory neurons [46,47]. At the glomeruli, the electric signals are further relayed to two specific classes of neurons, the PNs and LNs. The PNs housed in the AL project their axons to the protocerebrum [41] and from here on to the mushroom body (MB) and lateral horn (LH). Thus, the information from the PNs is transferred to these paired neural organs, which comprise higher centers of the fly’s brain [48,49], (Figure 3B).

### 2.1. Olfactory Sensory Neurons (OSNs)

The OSNs form the most important aspect of olfactory detection in *D. melanogaster*. Their numbers exhibit great variations between the larval and adult stages of the fly. In larvae there are 21 OSNs per DO, whereas the adult fruit flies house much greater number of neurons. Each antenna displays ~1300 OSNs, whereas the actual number is 120 in each maxillary palp. An individual OSN typically embodies a single Or, although a few multiple receptor OSNs are also present in *D. melanogaster* [44,51]. The OSNs expressing identical receptor gene/genes then project to a specific glomerulus [52,53], where they synapse onto PNs and LNs [54] for further processing of the olfactory cues. A few studies have suggested that the number of OSNs converging into glomeruli varies between 52–53 OSNs per glomerulus [55]. However, each glomeruli has its own unique neuronal composition and glomerular volume, which is a result of varying numbers of OSNs and uniglomerular PNs received by different sets of glomeruli [56]. Additionally, an increased number of glomeruli per antennal lobe has been reported in *D. melanogaster.* The number is now 58, out of which 51 are olfactory [57] and 7, i.e., VP1d, VP1l, VP1m [58], VP2, VP3 [59], and VP4 [60,61,62] are thermo/hygrosensory in nature. They house a total of ~2600 antennal lobe sensory neurons (ALSNs) including OSNs and thermo/hygrosensory neurons (T/HSNs) [63].

### 2.2. Odorant Receptors (Ors)

The odorant receptors (Ors) in *D. melanogaster* are membrane-associated proteins by nature. They are encoded and expressed by *Or* genes [64,65]. The seven transmembrane domains possessed by these proteins are quite distinct and have no homology to the vertebrate GPCRs or Ors [66,67,68]. There are 60 *Or* genes in *D. melanogaster*, which encode 62 Ors by the process of alternative splicing (Table 1) [69,70,71]. However, studies using transgenic reporter techniques have changed the scenario to a considerable effect. These studies have revealed that the count of genes exclusive for antennal Ors is 40, whereas 7 genes translate receptors specific to maxillary palps [34,41,52]. The remaining *Or* genes are involved in the encoding of larval Ors and are not detectably expressed in adult fruit flies. There are 25 larval Ors, of which 13 are specifically expressed in larvae [70,72,73]. Additionally, Ors exhibit a unique feature. The receptors expressed in the antennae and maxillary palps of *D. melanogaster* are exclusive to them and are not expressed in each other, as established by several algorithmic studies. These studies have identified a dyad element that promotes the expression of maxillary-palp-specific Ors and a motif that represses their expression in the antennae [74]. One more restrictive feature of Ors is that the expression of *Or* genes in morphologically distinct sensilla follows a set pattern [75]. The study of this pattern has revealed a strong correlation between the developmental pathways of the sensilla and the types of Ors they express [32].

It is well-established that a solo *Or* gene is translated in each OSN [41,78,79]. However, a few studies have revealed interesting facts about *Or* gene expression. Each *D. melanogaster* OSN expressing an Or also expresses a co-receptor named as Orco/Or83b [65,80]. This co-receptor is essential to the appropriate ciliary routing and functioning of every single Or [65,81]. The flies devoid of Orco exhibit defective behavioral and electrophysiological responses to a number of odorants [67]. Besides this, four populations of OSNs co-express two conventional Ors (Or33a/Or56a, Or33b/Or47a, Or33b/Or85a and Or33c/Or85e) and a fifth one co-expresses one Or and one Gr (Or10a/Gr10a) along with the co-receptor Orco leading to a modulated ligand response profile [51,52]. A couple of recent works using cryo-electron microscopy have further given an insight into the functioning of Or/Orco heteromeric complexes. Butterwick et al., through structural analyses, has found that an Orco homomer consists of four subunits organized in a symmetrical fashion around a central pore. Each Orco subunit is further composed of seven transmembrane helical segments (S1–S7), out of which S7a (the cytoplasmic section) forms the crux of the anchor domain, whereas S7b (the transmembrane section) contours the central pore. The pore is narrowest at the extracellular end of S7b due to the presence of hydrophobic residues Leu473 and Val469. When a ligand binds to the Or/Orco heteromeric complex, the hydrophobic aperture is expanded. This central ion-conduction pathway then veers into the four lateral channels (6-Å long) that open to the cytosol, providing an uninterrupted pathway for the transfer of ions [82]. On a similar note, Marmol et al. has deciphered that the odorant receptor MhOr5 in *Machilis hrabei* exhibits identical quadrivial structural organization and functions similarly to Orco. Here, the binding of the ligand also dilates the S7b helices to gate the ion conduction pathway. In nutshell, these structural understandings of the Or and Orco have shed a light on the promiscuous nature of these receptors, allowing insects to have a versatile chemical recognition system [83].

### 2.3. Ionotropic Receptors (Irs)

Besides Ors, an additional group of receptors, called ionotropic receptors (Irs), is also involved in olfaction in *D. melanogaster*. Structural analyses have revealed their similarities to the ionotropic glutamate receptors (iGluRs) with a homology of less than 34% [84,85]. They also share the ion-channel-like characteristic of iGluRs, though the binding site for glutamate is not present [86,87]. The chemosensory apparatus of the larval stage is mainly distributed to the head region, encompassing the dorsal organ (DO), terminal organ (TO), ventral organ (VO), dorsal pharyngeal organ (DPO), dorsal pharyngeal sensilla (DPS), ventral pharyngeal sensilla (VPS), and posterior pharyngeal sensilla (PPS), all existing in pairs [45,88,89,90]. All of the above organs express Irs. A few Irs are also present in the body area that forms the complete larval repertoire together with the receptors of the head region (Table 2) [91]. Contrarily, the distribution of Irs in adult flies is quite diverse (Table 3). In the head region, they are dispersed in the antenna, labral sense organ (LSO), dorsal cibarial sense organ (DCSO), ventral cibarial sense organ (VCSO), and the labellum. The Irs are also expressed in the wings and legs of the fruit fly [91].

The Irs confer odorant responses much like the Ors, but more in-depth studies need to be conducted in this direction. The number of *Ir* genes in the *D. melanogaster* genome is 61, in addition to 2 putative pseudogenes. Among these, 17 genes encode the antennal Irs, while the rest of them are translated and scattered all over the fly’s body (both in the larval and adult stages); the latter are called divergent Irs. Amid the antenna, out of 13 Irs, the majority of them sense olfactory cues from amines, aldehydes and evaporative acids, and few Irs are hygrosensory or thermosensory in nature [60,85,91,92]. The remaining four Irs, i.e., Ir8a, Ir25a, Ir76b, and Ir93a are present and function as co-receptors in different amalgamations [93,94]. This paired functioning of Irs with co-receptors suggests that they work in tandem, resulting in ion channels responding to volatile odorant molecules [94,95]. Additionally, the identification of the co-receptor extra loop (CREL), a conserved sequence of the Ir co-receptor ligand-binding domain (LBD) has brought an array of new insights. It has helped in understanding the assembly and stoichiometry of Ir complexes and their intracellular transport in a much-improved way [96]. Besides olfaction, the Irs are also involved in the perception of gustatory stimuli. A group of approximately 35 Irs, the Ir20a clade is translated in all of the taste organs of the fly co-expressed with bitter or sweet Grs [97]. This is explained in detail later, in the gustatory section of the review.

**Table 3 insects-13-00142-t003:** Ionotropic receptors (Irs) in *Drosophila melanogaster* (adult).

Antenna	LTB	LTP	LSO	VCSO	DCSO	Legs	Wings	Abdomen
Ir8a	Ir7a	Ir25a	Ir10a	Ir7a	Ir10a	Ir7a	Ir7a	Ir7a
Ir21a	Ir7c	Ir56d	Ir20a	Ir11a	Ir20a	Ir7c	Ir7b	Ir7c
Ir25a	Ir11a	Ir76b	Ir25a	Ir20a	Ir25a	Ir11a	Ir7c	Ir10a
Ir31a	Ir25a		Ir56a	Ir25a	Ir76b	Ir20a	Ir10a	Ir11a
Ir40a	Ir47a		Ir60b	Ir56a	Ir100a	Ir25a	Ir25a	Ir20a
Ir41a	Ir56a		Ir60c	Ir76b		Ir47a	Ir52a	Ir25a
Ir64a	Ir56b		Ir60d	Ir94a		Ir52a	Ir52b	Ir56a
Ir68a	Ir56d		Ir67c	Ir94b		Ir52b	Ir52c	Ir76b
Ir75a	Ir60c		Ir76b	Ir94c		Ir52c	Ir56a	
Ir75b	Ir60d		Ir94a	Ir94h		Ir52d	Ir56d	
Ir75c	Ir76b		Ir94e			Ir56a	Ir60c	
Ir75d	Ir94b		Ir94f			Ir56d	Ir76b	
Ir76a	Ir94e		Ir94h			Ir60c	Ir94e	
Ir76b			Ir100a			Ir67a		
Ir84a						Ir76b		
Ir92a						Ir94e		
Ir93a						Ir94h		

Outline of distribution of ionotropic receptors (Irs) in adult flies. (Adapted from [91,97]). LTB—Labellar taste bristles, LTP—Labellar taste pegs, LSO—Labral sense organ, VCSO—Ventral cibarial sense organ, DCSO—Dorsal cibarial sense organ.

### 2.4. Odorant Binding Proteins (Obps)

Encoded by a clan of 52 genes, the Obps are found in abundance in *D. melanogaster* [92]. They are a group of highly divergent pint-sized proteins of 13–28 kDa secreted in the olfactory sensillar lymph [98,99]. The Obps ease the transportation of hydrophobic odorants such as a few food odorants, pheromones, etc. to the Ors [98,100,101]. The Obp76a, also known as LUSH, is essential for the detection of the pheromone cis-vaccenyl acetate (cVA) by Or67d in trichoid sensilla [102]. However, various studies have shown that even in the absence of Obp76a, *D. melanogaster* can detect cVA, refuting its essentiality [103,104]. Another study involving Obp28a, one of the most abundant Obps in fruit flies, showed that it was not involved in the transportation of odorants [105]. Contrary to these findings, recent work has established that Obp28a is essential for the detection of the floral odorant β-ionone by fruit flies [106]. Thus, these findings of the functioning of different Obps suggest that a much larger picture of the exact role and mechanism of their operation remains to be deciphered.

## 3. Olfactory Signaling Pathway—Interpreting Olfaction in *Drosophila melanogaster*

### 3.1. Larva

The larvae and adults of *D. melanogaster* share a general outline of the olfactory system, though in larvae, it is smaller and simplified. The presence of a reduced number of OSNs in larvae suggests a diminished primary olfactory repertoire as compared to adult flies [107]. The 21 OSNs in the DO are connected to the LAL by the AN. The LAL consists of approximately 30 subunits showing resemblance to the adult glomerulus, though smaller in number and size [4,45]. In the LAL, the OSNs interact with 21 uniglomerular PNs, 14 multiglomerular PNs, 14 GABAergic and cholinergic LNs, 4 neuromodulatory neurons, 6 subesophageal zone (SEZ) neurons, and 1 descending neuron. The olfactory signals are thus transmitted to higher centers of the larval brain to elicit different behavioral responses [108,109].

### 3.2. Adult Fly

The antennae and maxillary palps are key to olfaction in adult fruit flies. They house several OSNs that are primarily involved in sensing odorants in *D. melanogaster* [110]. Studies have revealed that an Or behaves differently to different odorants. It can be excited by some odorants and inhibited by others [111]. Such a response shows the temporally complex nature of odorant signaling. Barring a few cases where the OSNs express either multiple Ors—such as ab5 sensilla, where Or33b and Or47a, are expressed together [51]—or receptors of the gustatory family [112,113], the majority of the 1300 OSNs in the *D. melanogaster* genome expresses one Or per neuron from the total of 61 Ors [36]. An additional Or, Or83b/Orco functioning as a co-receptor is also expressed in each neuron. It acts in concert with conventional Ors to recognize different odorants [65].

The point of emphasis is this: how do OSNs further process the olfactory stimuli they sense? The answer lies within the antennal lobe. Each of the 52 glomeruli it houses receives projections from the OSNs embodying the same Or [41,46,49,52,60]. From here on, the stimuli are relayed on to a network comprising projection neurons (PNs), inhibitory local neurons (iLNs), and excitatory local neurons (eLNs). The PNs are involved in transmitting olfactory information to higher centers of the fruit fly’s brain, i.e., MBs and LHs, whereas iLNs and eLNs establish lateral connections among different glomeruli [114]. What is interesting at this point is the systemization of the PNs into two different neural tracts. They are the:

**a. Inner antennocerebral tract (iACT)**: the calyx neuropil of the MB, along with the LH, forms synaptic connections with the PNs of the iACT [115].

**b. Medial antennocerebral tract (mACT)**: The PNs of the mACT bypass the MB calyx and go straight to the LH [116]. A few cases have revealed that part of the PNs of the mACT sends its extensions into multiple glomeruli, which trigger an inhibitory response.

This is how olfactory stimuli are perceived by the OSNs and relayed to the higher centers of the brain for further processing, to initiate varying behavioral responses in adult fruit flies.

## 4. Molecular Basis of Olfactory Signaling—Decoding the Game of Smell

In *D. melanogaster*, the Or targeting a specific odorant (OrX) forms a heteromeric complex with the universal co-receptor Orco [117]. The mode of action of this complex is synonymous with the functioning of a ligand-activated cation channel [118,119]. On the other hand, the Irs are known to form heteromeric complexes with up to three different subunits. They consist of a variable, odorant-specific Ir (IrX), and one or two broadly expressed co-receptors out of Ir8a, Ir25a, and Ir76b [93,94]. The Irs exhibit a distinct similarity with iGluRs in having a transmembrane domain, a ligand-binding domain (LBD), and an aperture region [120]. Various models have been extensively studied focusing on the functioning of the above-mentioned olfactory receptors in *D. melanogaster*. These models include the molecular signaling pathways involved in the sensing of food odorants, pheromones, and CO_2_, respectively. To have a much better understanding of the cited models, let us divulge the details of these signaling cascades.

### 4.1. General Odorant Sensing

The odorant-binding proteins (Obps) carry odorant molecules to the OrX-Orco heteromeric complex. The binding of the ligand and receptor leads to the activation of an ion channel [121]. This channel is a non-specific cation channel permeable to Ca^2+^ and other positively charged ions (Na^+^, K^+^). The permeability leads to an influx of cations changing the membrane potential of the OSNs, leading to their depolarization and repolarization in a subsequent manner [118]. Thus, the olfactory signal (electrical activity) is transmitted along the OSNs to the PNs and then to higher centers of *D. melanogaster’s* brain. This type of signaling is known as ionotropic signaling (Figure 4). In parallel, there are several reports that point towards the involvement of G-proteins in odorant detection. It was found that the olfactory acumen of flies was altered by mutations targeting the cAMP signaling cascade [122]. The Ors expressed in HEK293 cells, when stimulated by an odorant, resulted in an enhanced cAMP production [123]. A decreased level of cAMP was also reported to alter the proper detection of odorants by flies [124], whereas odorant stimulation led to an increase in cAMP production [125]. In addition, the role of Gαq [126], Gαo [127], and Gαs in the transmission of odorant stimuli further characterized the involvement of phospholipid and cAMP second messengers in olfactory signaling [128]. Still, the involvement of G-proteins in odorant detection is still debatable now [129] and is most likely to modulate the functioning of ion channels in one or another way.

### 4.2. Signaling in Pheromone-Sensing Ors (Prs)

The pheromone molecules attached to the pheromone-binding proteins (PBPs) are fixed to the PrX-Orco heteromeric complex. The SNMP1, i.e., sensory neuron membrane protein 1 most likely associates with the complex and mediates the transfer of pheromones to PrX (pheromone specific Or protein) [131]. The pheromone-receptor complex then activates an ionotropic channel (PrX-Orco channel), leading to an influx of cations (Ca^2+^, Na^+^, K^+^) and subsequent depolarization of the pheromone sensory neurons (PSNs) [132]. This results in the transmission of the olfactory signal (Figure 5).

### 4.3. Signaling in the Sensing of CO_2_

The CO_2_ triggers a strong and inborn evading olfactory behavior in *D. melanogaster* [133,134]. Its sensing at a concentration of less than ~2% is mediated by the class of OSNs (ab1C) co-expressing Gr21a and Gr63a [133,135,136,137], whereas detection at a higher concentration of more than 5% involves acid-mediating Ir64a and Ir8a [92,138]. A few reports propose that at lower concentrations, the G-proteins might be involved in the detection of CO_2_ by fruit flies. The Gαq and Gγ30A were found to play a role in the signaling of CO_2_ in *D. melanogaster* [129]. Another work hinted that the CO_2_ receptors Gr21a and Gr63a might activate the TRPC channels through Gαq and PLC21C [139]. However, this is not an established view and more work needs to be conducted in this direction to further validate the involvement of G-proteins in CO_2_ signaling. At higher levels of CO_2_, Ir64a and Ir8a act in tandem to decipher a dip in the pH of the sensory lymph to trigger an avoidance behavior [92,138]. Besides this, a different study has put forward the participation of PNs innervating the V-glomerulus (PN_v_s) in managing the avoidance response of fruit flies to low and high CO_2_ concentrations. It was shown that the PNv-1pathway is activated at lower concentrations of CO_2_ (0.5%), whereas higher levels of CO_2_ (2%) stimulate PNv-1, PNv-2, and GABAergic PNv-3 pathways [140]. Another interesting finding involving the sensing of CO_2_ by *D. melanogaster* has come to the light. It has been found that fruit flies, while walking, perceive CO_2_ as an aversive stimulus, whereas the same stimulus while flying becomes attractive. This in-flight attraction to CO_2_ involves different components working in tandem, i.e., Ir64a, Orco, and octopamine signaling [141]. However, recent work has put forward the fact that fruit flies perceive CO_2_ as an attractant in both walking and flying conditions when they are in an activated stage of searching for food. This modus operandi is facilitated by the functioning of a distinct chemosensory pathway involving Ir25a [142].

In nutshell, these models highlight the functioning of insect olfactory receptors as heteromeric ligand-gated ion channels. Novel research decoding the structure of Orco has further substantiated the involvement of ion conduction pathways in the perception of olfactory stimuli by insects [82].

## 5. Olfaction-Based Repellency—The Established and New Players on the Horizon

The process of olfaction is indispensable to the survival of insects. It allows them to detect food, probable mates, and the signs of danger so as to avoid them. Likewise, the disease-transmitting insect vectors also use their sense of smell to trace and feed on their human hosts [143,144,145,146]. Additionally, crop infesting insects such as *Drosophila suzukii*, a major global fruit pest, use olfactory detection of higher CO_2_ emission to target ripening fruits [147]. This results in a humongous amount of agricultural loss globally, accounting for millions of dollars [147,148,149]. Consequently, targeting and modulating the olfactory system has become a prominent choice for researchers to control the menace of insects and develop better alternatives for combating insect pests.

Insect repellents are still the first line of defense against disease-causing insect vectors. The major ingredient used in these current batches of repellents is N, N-diethyl-m-toluamide (DEET), which triggers an airborne repugnancy among insects [146,150,151]. As a result, DEET is wildly used to ward off blood-sucking mosquitoes to limit the transmission of pathogens among humans [150,152]. To understand the molecular machinery behind this olfaction-based repellency caused by DEET, many researchers have opted for *D. melanogaster* as a model organism. In one of the research works, it was observed that fruit flies with intact antennae avoided DEET-treated food vials. Conversely, flies devoid of both of the antennae, with their maxillary palps unharmed, did not show any repellency to DEET. Thus, it was concluded that the DEET repellency is olfaction driven as antennae house OSNs. To confirm the findings, Or83b/Orco mutant fruit flies were used, as Orco is an indispensable co-receptor required for the proper functioning of Ors [65,67]. The results were consistent, as wild-type flies avoided DEET-treated food vials while Orco mutants were insensitive to DEET. Further in-depth analysis revealed that the reduced acuity of food odorant was related to the electrophysiological inhibition of Or82a/Orco housed in the ab5A sensillum and Or47a/Orco in ab5B sensillum. In summary, it was concluded that DEET reduces the perception of food aroma in fruit flies by inhibiting the odorant-induced activation of a subset of Or/Orco complexes to varying degrees [152]. Similarly, using Orco mutant mosquitoes exhibited that they were not repelled by volatile DEET, indicating the need for an intact Or pathway for DEET repellency [151].

A different work using *D. melanogaster* also exhibited involvement of the olfactory system in the functioning of DEET. The study revealed that Or42a was involved in the DEET repellency. The receptor was also sensitive to two other major repellents—picaridin and Ir3535—hinting towards its generic nature [25]. However, a recent analysis with fruit flies has unveiled that, although higher concentrations of DEET were detected by the Orco based olfactory system, there was no specific OSN involved in this process. The deletion of Or42a and Or2a neurons showed no difference in DEET avoidance, pointing towards the involvement of multiple OSNs in the repugnancy of DEET [153]. Altogether, it can be inferred that the repugnant behavior of insects towards DEET involves their olfactory system. Still, there are several conflicting models involved in DEET olfactory repellency. DEET is found to work by masking odorants. One such example is the reduced attraction of *Aedes aegypti* mosquitoes to humans by masking the smell of lactic acid [154]. DEET is also known to cause a dip in the release of 1-octen-3-ol, disrupting the host-seeking mechanism of insects by changing the chemical profile of the host’s skin odorants [155,156]. Another hypothesis of the functioning of DEET is that it acts as a confusant. It is likely to alter the regular activation pattern of the glomeruli of insects, thereby blocking the host’s attractiveness. Besides this, as per the smell and avoid model, DEET activates aversive GRNs in insects that override the neural activity of GRNs sensing attractive cues coming from the host. This allow insects to preferably avoid their hosts [157]. In addition to DEET, pyrethrum is also an extremely popular insect repellent that has been used by humans for a long time to control arthropod pests. It is extracted from the dry flowers of the plant *Tanacetum cinerariifolium*. However, the molecular mode of functioning of this repellent was still a question that needed to be answered. Currently, it has been revealed that there is the involvement of Or7a, Or42b, Or59b, and Or98a in avoidance behavior towards pyrethrum. The first three receptors mentioned above are activated by pyrethrins which are the key insecticidal constituents of pyrethrum, whereas Or98a is stimulated by E-β-farnesene (EBF), a minor constituent. It was observed that genetically knocking out Or7a, Or59b, and Or98a obliterated the repellency of fruit flies to pyrethrum, pointing towards their involvement in avoidance behavior [158].

DEET is a highly efficient insect repellent, with its fair share of problems. First, it needs to be applied on a recurrent basis in increased concentrations making it quite expensive for the people in areas infested with vector-borne diseases [150]. In addition, a reduced repugnancy by DEET is observed in mosquitoes owing to its repeated exposure. Therefore, there have been efforts from the research community to design and develop better alternatives to DEET, to deal with the menace of insect vectors and pests. A few of the prominent examples are as follows:

**a. Geosmin:** Trans-1,10-dimethyl-trans-9-decalol, commonly known as Geosmin is a compound with an earthy smell produced by a few specific groups of bacteria, fungi, and cyanobacteria. Using *D. melanogaster* as a model organism, it was revealed that at extremely low concentrations, Geosmin triggered repellency. Further in-depth analysis exhibited that this aversive behavior was triggered by Or56a housed in the ab4B OSNs [156,159]. Geosmin is also known to induce a dip in the attraction of fruit flies towards vinegar compounds [159,160]. Thus, the ability of Geosmin to induce aversion and modulation of inborn attraction makes it a very desirable candidate for a strong repellent.

**b. OX1w:** In insects, each OSN expressing an Or also expresses a co-receptor named Orco [65,80]. This co-receptor is indispensable to the proper ciliary routing and functioning of every single Or [65,81]. In *D. melanogaster*, the attraction towards ethyl acetate (EA) is mediated by the Or42b/Orco heteromeric complex [161]. A group of researchers working on fruit fly larvae found that the air-borne application of the Orco antagonist, OX1w, eliminated their chemotactic movement towards EA [146]. Thus, inhibiting the functioning of Orco by designing more potent Orco antagonists can be a potential method to control insect vectors and pests.

**c. Coffee furanone:** 2-methyltetrahydrofuran-3-one, also known as coffee furanone is a natural volatile compound, found to have repellent/attractant properties. A recent work using *D. melanogaster* showed that the application of coffee furanone triggered an aversive behavior in fruit flies. An in-depth analysis brought to light the functions of this compound, as activating the variable OrX subunit of the OrX/Orco heteromeric complex is necessary for odorant detection. What makes coffee furanone special is its promiscuous nature, as it is known to stimulate nearly 80% of the OSNs in the antennae of *D. melanogaster* and various other insect species. In addition, coffee furanone is a common flavoring agent used in food and is considered safe for humans. Consequently, all of these features of coffee furanone brands it a highly desirable candidate to serve as a repellent or attractant to combat insect vectors or crop pests [121].

**d. Essential oils:** Since it has already been shown that insects use their olfactory system to detect food and hosts, the use of plant-based essential oils to repel them has immense potential to be developed into high-quality repellents. Several essential oils such as camphor, peppermint oil, norcamphor, nerol, menthyl acetate, menthone, (-)-menthol, etc. have been found to elicit repellency in *D. melanogaster* and *D. suzukii*, which are serious global crop pests. It was observed that aversion to essential oils was reduced in Orco mutant fruit flies, pointing towards their olfactory mode of functioning. Thus, further in-depth study of the molecular mechanism of the operation of essential oils will be helpful in the long run to ward off the ever-increasing menace of the insects [162] (Figure 6).

## 6. Gustatory System—Components and Basic Organization

Unlike mammals, the gustatory system in fruit flies is distributed throughout their bodies. The taste bristles encompassing the taste neurons in an adult *D. melanogaster* are present in the proboscis, legs, ovipositor, and wing margins of the fly [163,164,165]. However, in the larval stages, the taste system is limited to the head and pharynx regions [12,166]. The dorsal (DO, peripheral region), terminal (TO) and ventral organs (VO), along with the dorsal (DPS), ventral (VPS) and posterior pharyngeal sensilla (PPS), and the dorsal pharyngeal organ (DPO), constitute the larval gustatory system (Figure 7A) [89,167]. These organs of appetence exist in sets of two, consisting of varied numbers of sensory neurons organized in sensilla. The DO has 11 neurons distributed in 6 peripheral sensilla, the TO has 37 neurons in 17 sensilla, the VO has 7 neurons in 5 sensilla, the DPS has 16 neurons in 6 sensilla, the VPS has 17 neurons in 4 sensilla, the DPO has 5 neurons, and the PPS has 6 neurons in 2 sensilla each [168]. In adult fruit flies, the taste organs comprising the mouth part are both extrinsic and intrinsic in nature. The extrinsic taste organs composed of the labial palps are connected to the proboscis, the tubular mouthpart used for feeding. When *D. melanogaster* is indulged in active feeding, its labellum opens up to allow the food item to interact with the taste pegs, followed by the food’s entry into the pharynx and esophagus [169]. Here, the intrinsic taste organs are involved. These three organs, namely the labral sense organ (LSO), ventral cibarial sense organ (VCSO), and dorsal cibarial sense organ (DCSO), are found in the interior wall of the pharynx [170]. They consist of tightly packed cell clusters and are involved in the monitoring of ingested food (Figure 7B). The internal taste organs help *D. melanogaster* in deciding whether to perform a regurgitate response to a harmful substance or to enhance sucking reflexes to a desirable food [171].

Looking closely at the elementary architecture and distribution of taste sensilla in fruit flies, a terminal pore is seen at the tip of the taste bristles and pegs (Figure 8A). When fruit flies ingest any food item, it contacts dendritic processes of the gustatory receptor neurons (GRNs). The dendrites extend into the bristle shaft and may help in further processing of the gustatory information [169,173]. Additionally, amid the inner surface of the bristle and dendrites, there is a fluid-filled space. This fluid is called lymph, and is believed to make food tastants accessible to their related receptors [174]. The taste lymph is secreted by the support cells, though little is known about its basic composition. The legs and anterior wing margin of *D. melanogaster* also house some taste bristles [13]. This wide distribution of gustatory hairs serves as survival kit for fruit flies (Figure 8B). They can analyze the nature of the food, and assess whether it is toxic or nutritious before extending their proboscis for ingestion [175]. In females, a few sensilla are also present around the ovipositor that might help *D. melanogaster* in finding places that have enough nutrient availability for proper laying and conditioning of eggs [163,176,177]. However, recent work targeting complete anatomical, physiological, and molecular characterization of the ovipositor’s sensilla and pegs in *D. suzukii*, *D. melanogaster*, along with a few related species, has brought to light some interesting facts. The study through GAL4-driven expression of GFP in *D. melanogaster* showed that the pegs in the ovipositor house mechanoreceptors rather than chemoreceptors. This highlights a novel finding that the endpoint of the ovipositor in *D. melanogaster* is not involved in the assessment of nutritional content of the egg-laying substrate [178]. Besides this, the involvement of taste bristles in courtship and mating has also been shown. A few studies suggest that male flies use sensilla on their forelegs to detect pheromones, promoting copulation with females [19,179]. Thus, gustatory sensilla are indispensable to the existence and survival of fruit flies.

The gustatory sensilla on the labellum can be classified into three groups based on their lengths (Figure 8C). They are the:

**a. Long-type (L-type):** The L-type bristles are long and harbor 4 GRNs per sensillum. These neurons are A, C, D_L_, and E cells, responding to sugar, water, high salt, and low salt concentrations, respectively [180,181].

**b. Intermediate-type (I-type):** The I-type gustatory hairs are intermediate in size. They house 2 GRNs per sensillum. Out of these two, one GRN (A) helps flies to detect sugar and low salt concentrations, while the other one (B) mediates the response to high salt concentrations and aversive compounds [17,182,183]. The I-type sensilla are narrowly tuned and include two sub-classes, I-a and I-b, that are distributed on the lateral regions of the labellum in the anterior and posterior regions, respectively [17].

**c. Small-type (S-type):** As the name suggests, the S-type bristles are small in size. Like the L-type, they also house 4 GRNs per sensillum, i.e., A, B, C, and D_S_ cells, showing gustatory responses to sweetness, bitterness, water, and high salt concentrations, respectively [180,183]. The S-type sensilla are further divided into three sub-types, i.e., S-a, S-b, and S-c, amalgamated with each other on the medial region of the labellum. Out of these three, S-a and S-b are the most broadly tuned sensilla, with S-b showing a much greater response to the tastants [17].

In addition to GRNs, the gustatory sensilla in *D. melanogaster* also house mechanosensory neurons (MNs). As a result, when fruit flies feed, the taste sensilla receive both gustatory stimuli as well as the mechanical sensation [91,184]. Thus, the gustatory coding in the legs of fruit flies also needs to be investigated in detail to understand the complete repertoire of organs involved in taste perception. The tarsal segments of the legs have a lot more sensilla as compared to labellum, and were found to be the key in the sensing of gustatory stimuli. It is the tarsi that come in initial contact with food items, followed by the labellum [185]. In female *D. melanogaster*, gustatory sensilla are distributed on the five tarsal segments of the forelegs, midlegs, and hindlegs of the fly in amounts of ~28, 21, and 22, respectively, expressing 28 Grs in various combinations. These sensilla can be divided into four functional groups, which are as follows:

**a. Type A1:** Specific to the forelegs only, the Type A1 includes f5s, the most broadly tuned sensillum, responding to a wide range of sweet and bitter compounds such as sucrose, maltose, spartein, denatonium, etc.

**b. Type A2:** Comprising of f4s and f5b sensilla, they respond to a much narrower range of bitter tastants, though the sugar reception profile is similar to the Type A1.

**c. Type B:** The repertoire consists of the sensilla f2b, f3b, and f4b. These sensory hairs show a negligible response to the bitter compounds, and sugar taste reception is also on the lower side as compared to Type A1 and A2.

**d. Type C:** The sensilla in this functional group consist of f3a, f1a, f5a, f1c, f1d, and f2a. These gustatory hairs exhibit extremely weak responses to both sweet and bitter compounds [186].

Further, two more types of foreleg sensilla, namely f5v and f4c were identified. The f5v sensillum exhibits a strong response to sweet tastants, but reacts poorly to bitter compounds. Contrarily, the f4c sensillum responds to a wide array of bitter compounds such as lobeline, denatonium, etc. with fairly good measure. The gustatory sensillar organization in the mid and hindlegs of the female fly is also quite similar to the foreleg, barring a few exceptions. A limited number of sensilla, namely f5s, f4c, f1b, and f3b, are only present in the foreleg; furthermore, the f3a sensillum of the midleg is not paired. Besides this, in male *D. melanogaster*, the distribution of the gustatory hairs is comparable to female flies, with only forelegs showing some additional taste sensilla [186].

The GRNs in the gustatory sensilla extend their axons to the subesophageal ganglion (SEZ) [187]. It is widely regarded as the primary taste center in *D. melanogaster* [188]. The SEZ is located in the ventral area of *D. melanogaster’s* brain and plays an important role in the perception and processing of taste stimuli [189,190]. Each region of SEZ responds to different types of taste. The anterior and dorsolateral regions respond to sweet/bitter stimuli from the pharyngeal GRNs, whereas the posterior SEZ retort to the tarsal’s sweet/bitter taste neurons. In addition, the labellar bitter/sweet neurons send their projections to the medial SEZ [191].

### 6.1. Gustatory Receptors (Grs)

The Ors and Grs in mammals are prime examples of typical G-protein-coupled receptors (GPCRs). These receptors are transmembrane in nature and share a seven hydrophobic domain structure consisting of α-helical chains [192]. However, in *D. melanogaster* the Grs are not related to those in mammals, and are not GPCRs in nature. Indeed, in fruit flies, the Grs are evolutionarily linked with Ors [66,69,164,179,193,194,195,196,197]. Further, several studies substantiate that like Ors, the Grs are also ligand-gated cation channels, which points towards a close relationship between these two receptor families [14]. Presently, the number of Grs in *D. melanogaster* is 68 (encoded by 60 *Gr* genes), owing to the completion of the *D. melanogaster* genome project [69]. Apart from their roles in the detection of tastes, a few Grs have also been found to be expressed in OSNs [112,170]. The Gr21a and Gr63a co-expressing OSNs, involved in the detection of CO_2_, are prime examples of this [198].

As far as the larval stages of *D. melanogaster* are concerned, the translational assay of the Gr genes employing Gr-Gal4 lines has exhibited some interesting facts. It has shown that 39 Grs are expressed in the TO, DPS, VPS, or PPS of larva, comprising the larval gustatory system [167,199,200] (Table 4). Most larval Gr-GAL4s also exhibit co-expression alongside Gr33a-GAL4 and Gr66a-GAL4. This pattern of countenance advocates that most larval Grs might respond to a bitter taste [167]. One more important thing which has come to light from Gr-Gal4 expression analysis is that *D. melanogaster* larvae do not express major receptors needed for sugar sensing. This absenteeism of the sugar Grs (Gr5a, Gr61a, and Gr64a-f) raises an important question. The underlying question is: how do larvae respond to sugars? The answer lies within Gr43a, the fructose receptor that might trigger sugar detection in larvae [168,201].

The distribution of Grs in adult fruit flies is diverse. Out of 68 Grs known, the expression analyses of the majority of them have revealed that 53 Grs are expressed in the labellum, 32 Grs in the tarsal leg segments, 19 Grs in the taste pegs and pharynx, 7 Grs in the wing margins, and 5 Grs in the antenna [191] (Table 5). This vast repertoire of Grs allows *D. melanogaster* to respond to an array of tastants in an efficient manner. The taste receptors are classified into different classes:

**a. Bitter taste receptors:** The structure and function of the bitter taste receptors in *D. melanogaster* are more complex than their mammalian counterparts, consisting primarily of homo- and hetero-dimers [16,202]. A few research works in this direction have identified the involvement of multiple Grs in fruit flies for triggering a response to various bitter tastants. The co-expression of Gr66a, Gr93a, and Gr33a in a single GRN, along with their inevitability for responding to caffeine, hints towards a multimeric architectural ensemble of the bitter taste receptors [15,16,203,204]. The response profile of various other bitter tastants has further substantiated this finding. The hetero-multimers of Gr32a/Gr59c/Gr66a or Gr22e/Gr32a/Gr66a are needed for the detection of denatonium (DEN), lobeline (LOB), and berberine (BER), while Gr22e/Gr32a/Gr66a responds additionally to strychnine (STR). Besides this, Gr32a/Gr33a/Gr66a retorts to quinine (QUI), denatonium (DEN), escin (ESC), sparteine (SPS), berberine BER), and lobeline (LOB) while responses to theobromine (THE), umbelliferone (UMB), caffeine (CAF), and theophylline (TPH) necessitate the four-receptor complex of Gr33a/Gr39a/Gr66a/Gr93a [20,205].

It has been found that, among the complete repertoire of the bitter Grs, there is a ubiquitous expression of Gr32a, Gr33a, Gr39a.a, Gr66a, Gr89a, and Gr93a in all bitter GRNs designated as “core bitter Grs” or “commonly expressed receptors” (CERs). These core Grs may function as co-receptors (analogous to Or83b/Orco) with other Grs further reinforcing their multimeric organization and enhanced complexity [17,19,20]. In addition to bitter taste receptors, the transient receptor potential (TRP) channels were also involved in aversive taste stimulation. The GRNs in the labellum express three TRP channels—TRPA1, Painless, and TRP-Like (TRPL)—showing behavioral avoidance to aristolochic acid (AA) [206], isothiocyanates [207], and camphor, respectively [208]. As far as aristolochic acid is concerned, at higher concentrations, it is sensed directly by TRPA1, whereas at lower levels, it is picked up through a signaling cascade initiated by opsins that later on couples to TRPA1 [209].

**b. Sugar taste receptors:** The sugar-responsive GRNs such as bitter neurons express multiple Grs in a combinatorial fashion. This points towards their dimeric or multimeric mode of functioning [210,211,212,213]. A common Gr found in most of the GRNs that are stimulated by a small subset of sugars, particularly trehalose, is Gr5a [210,214,215,216]. It has been deciphered that the Gr5a-positive taste neurons also express Gr61a and a group of six closely related Grs (Gr64a, Gr64b, Gr64c, Gr64d, Gr64e, and Gr64f) in a tandem array, designated as the “candidate sugar receptors” [69,210,211,213,217]. To elaborate, the sugar Grs in *D. melanogaster*, based on combinatorial expression in labial palps and tarsi, exhibited varied patterns of translation. First, 7 out of 31 sensilla in each palp express all sugar Grs except Gr64a. Second, 6 S-type sensilla translate Gr64f and Gr5a, along with Gr64b, Gr64c, and Gr64e in a random fashion. Third, 6 neurons linked with the taste pegs exhibit strong co-expression of Gr5a and Gr64c in addition to Gr64f. Finally, 4 neurons in the LSO display the presence of Gr61a, Gr64a and Gr64b in addition to Gr64f. Besides this, the tarsal sweet Grs also display different outlines of neuronal co-expression. The sensilla f5s and f4b express all of the candidate sugar Grs, and f5a sensilla express Gr61a, Gr64b and Gr64f, whereas f5v sensilla express Gr61a, Gr64a, Gr64c, Gr64f, and Gr43a [213].

Among the Gr64 cluster, Gr64a, along with Gr5a, are of prime importance for the gustatory behavior of *D. melanogaster*. The Gr5a is tuned to a narrow range of sugars such as trehalose, m-a-glucoside, glucose and melezitose, whereas Gr64a acts as a receptor for a wide range of sugars such as sucrose, maltose, turanose, raffinose, palatinose, maltotriose, leucrose, stachyose, maltitol, and fructose [210]. A study using Gr64 mutant flies (devoid of Gr64a-Gr64f) has thrown light on the multimeric operation of sugar Grs. It has been found that Gr64 mutant flies exhibited a highly reduced response to trehalose, even in the presence of a functional Gr5a gene. It underlines the fact that the tracking of trehalose by *D. melanogaster* requires the functioning of two or more Grs as a dimeric or multimeric unit [212]. This finding has been further validated by another study using Gr64 mutant flies. The study has revealed that Gr5a and Gr64a require Gr64f as a co-receptor forming a functional heterodimer for normal trehalose and sucrose detection, respectively, in fruit flies [218]. Besides this, the receptors for attractants other than sugars are not well established until now, but a few works in this direction have unfolded the possible involvement of Gr64e in detecting fatty acids [219].

**Table 5 insects-13-00142-t005:** Gustatory receptors (Grs) in *Drosophila melanogaster* (adult).

Antenna	Labellum	Taste Pegs	Pharynx	Wings	Tarsal Leg Segments
Gr5a	Gr5a	Gr5a	Gr2a	Gr22a	Gr5a
Gr10a	Gr8a	Gr64c	Gr22b	Gr22e	Gr8a
Gr21a	Gr21a	Gr64f	Gr22c	Gr28b.c	Gr22a
Gr22e	Gr22a		Gr22e	Gr39a.d	Gr22b
Gr63a	Gr22b		Gr28a	Gr43a	Gr22c
Gr64b	Gr22d		Gr28b.a	Gr59c	Gr22d
Gr64f	Gr22e		Gr28b.c	Gr68a	Gr22e
Gr68a	Gr22f		Gr28b.d		Gr28a
	Gr23a.a		Gr28b.e		Gr28b.a
	Gr23a.b		Gr32a		Gr28b.c
	Gr28a		Gr33a		Gr28b.d
	Gr28b.a		Gr47a		Gr28b.e
	Gr28b.c		Gr64a		Gr32a
	Gr28b.d		Gr64e		Gr33a
	Gr28b.e		Gr66a		Gr36a
	Gr32a		Gr93a		Gr39a.a
	Gr33a				Gr39b
	Gr36a				Gr43a
	Gr36b				Gr57a
	Gr36c				Gr58c
	Gr39a.a				Gr59a
	Gr39a.b				Gr59d
	Gr39a.c				Gr61a
	Gr39a.d				Gr64c
	Gr39b				Gr64e
	Gr43a				Gr64f
	Gr47a				Gr66a
	Gr57a				Gr68a
	Gr58a				Gr89a
	Gr58b				Gr93a
	Gr58c				Gr93b
	Gr59a				Gr98d
	Gr59b				
	Gr59c				
	Gr59d				
	Gr59f				
	Gr61a				
	Gr64a				
	Gr64b				
	Gr64c				
	Gr64d				
	Gr64e				
	Gr64f				
	Gr66a				
	Gr68a				
	Gr89a				
	Gr92a				
	Gr93a				
	Gr93b				
	Gr98a				
	Gr98b				
	Gr98c				
	Gr98d				

Outline of distribution of gustatory receptors (Grs) in adult flies. (Adapted from [17,98,186,191,213]).

**c. Salt taste receptors:** The salts play a key role in the survival of organisms by regulating many important physiological processes. However, their effects change drastically with concentrations. A moderate salt concentration is beneficial (up to 100 mM), while a high concentration (more than 200 mM) have exhibited deleterious effects [220]. Thus, like many other organisms, *D. melanogaster* also displays a preference for low salt conditions. In the larval stage, two epithelial sodium (DEG/ENaC) channels are involved in the sensing of salts. These channels, namely PPK11 and PPK19, are expressed in the TO and trigger appetitive and aversive behavioral responses to low and high salt concentrations, respectively [221]. In addition, it has been found that besides PPK19, Sano, a cytoplasmic protein co-expressed with Gr66a in the TO, is also needed by larvae for detecting increased concentrations of NaCl [222]. Contrary to this, the sensing of salts in adult *D. melanogaster* is different and diverse. Here, in place of ENaC channels, the Ir76b is needed for low and high salt sensing [220,223]. However, a recent study has revealed that a wide array of receptors expressed in different classes of GRNs are also involved in salt detection. The sweet taste receptors Gr64f and Ir94e act as the two low salt cell type receptors, responding to lower levels of NaCl, whereas the bitter taste receptor Gr66a responds to higher concentrations of NaCl and KCl, behaving as a high-salt cell type receptor. Besides Gr66a, the PPK23^glut^ also acts as a robust high-salt cell type receptor triggering a strong avoidance behavior to various salty tastants such as KCl, CaCl, CsCl, NaBr, etc. [224].

**d. Sour taste receptors:** Many factors influence the choice of food that *D. melanogaster* makes, with acid content being one of them. Fruit flies prefer food items that are low in acid, while rejecting highly acidic compounds. Some reports suggest the role of bitter GRNs and Irs in the detection of acids. A subsection of neurons for bitter tastants detect carboxylic acid [225], whereas Ir7a perceives high levels of acetic acid [226]. As per a recent study, the tracking of low and high acidic contents by *D. melanogaster* involves two independent pathways. These pathways compete with each other to finalize the triggering of an appetitive or aversive response. Additionally, the role of Otopetrin-like a (OtopLa), a protein confined to the tips of a new class of GRNs in acid-sensing, has been discovered. It was observed that the OtopLa functions as a cation channel, and is involved in the attractive sensing of low levels of acids [227]. However, another new report has also discovered the essentiality of OtopLa in eliciting a repulsive behavioral response to food with a high acid content [228].

### 6.2. Ionotropic Receptors (Irs)

In addition to olfaction, the Irs in *D. melanogaster* are intricately involved in the gustatory system as well. It has been found that Irs, belonging to the clade Ir20a, perceive taste stimuli in both larvae and adult fruit flies [90,97]. In larvae, 11 Irs are known to be expressed in different gustatory organs. Out of this, Ir48c, Ir60b, Ir60e, Ir67b, Ir67c, Ir94h, and Ir94f are translated in the DPS, whereas Ir20a is manifested in the DPO. Additionally, the Ir47a is expressed in the TO in addition to the manifestation of Ir56a, Ir67b, and Ir94d in larval body. Thus, this distribution of Irs, particularly in DPS, hints at their major role in deciphering the metabolic compounds produced in the larval digestive tract. Besides the Ir20a clade, Ir76b and Ir25a from another class of Irs have also been found to be expressed in all of the larval gustatory organs [90]. In adult fruit flies, the distribution of the Ir20a clade is a bit more diverse. Here, ~35 Irs from this group are expressed in GRNs, either independently or co-expressed with the sweet and bitter taste receptors. However, it has been observed that out of 35 *Ir20a* genes, only 28 genes code for proteins, while the other 7 genes might be pseudogenes. Thus, out of these 28 Irs, Ir47a, Ir56a, Ir56b, Ir56d, and Ir94e are expressed in the labellar taste sensilla. It has also been found that the Ir56a is co-expressed with the bitter receptor Gr66a, whereas Ir56b and Ir56d exhibit co-expression with the sugar receptor Gr5a. This pattern of translation indicates their role in the detection of appetitive and aversive tastants individually. Furthermore, the members of the Ir20a group are also translated in the gustatory sensilla of *D. melanogaster* legs. The Ir52a and Ir56d are expressed in both the tibia and tarsal segments, whereas Ir20a, Ir47a, Ir52a, Ir52c, Ir52d, Ir56b, Ir62a, and Ir94b are translated solely in the tarsi. This distribution of Irs does not end here, but is further extended to the pharynx of the fly. The Ir20a, Ir56a, Ir60b, Ir67c, Ir94f, and Ir94h are expressed in the LSO, while VCSO exhibits the translation of Ir20a, Ir94a, Ir94c, and Ir94h. Lastly, Ir52a is expressed in the anterior margin of wings, thereby concluding the distribution of Irs in the gustatory system of the fruit fly [90,91,97]. In nutshell, this widespread distribution of the Ir20a group in gustatory organs, in addition to their axonal projections to the SEZ, points to their intricate role in the perception of taste stimuli [97].

### 6.3. Transient Receptors Potential Channels (TRP)

Along with Grs, TRP channels are also involved in the detection of aversive compounds by *D. melanogaster*. The roles of three TRP channels, i.e., painless, TRPA1, and TRPL, have been ascertained in this regard [206,207,208,229]. It has been found that TRPA1 is needed in a subset of bitter GRNs to trigger an avoidance response to aristolochic acid [206]. The role of TRPA1 and painless in the sensing of reactive electrophiles (a group of noxious compounds) such as allyl isothiocyanate (the colorless oil of wasabi) has also come to light [207,229]. Furthermore, TRPL is involved in the detection of unpalatable food items, such as camphor, by *D. melanogaster* [208].

## 7. Gustatory Signaling Pathway—Understanding the Chronology of Taste

The cascade of events involved in the perception of taste in *D. melanogaster* is not well recognized. It demands a more thorough understanding and in-depth study of the various factors that help GRNs in eliciting an action potential.

### 7.1. Detection of Sugary/Bitter Tastants

As mentioned earlier, the Grs in fruit flies exhibit an inverted topology, similarly to Ors, and function as ligand-gated cation channels [230]. This ionotropic mode of functioning has been further substantiated by a work involving Gr43a, a highly conserved receptor. This receptor bestowed the sensing of fructose onto extracted patches of membranes of Gr43a, expressing cultured cells [14,177]. However, altered detection of sweet tastants has been observed in mutant fly strains with distorted G-protein signaling [215,216,231,232]. In *D. melanogaster*, 11 genes have been identified until now that encode these G-protein subunits. There are six genes translating α subunits, whereas β and γ subunits are expressed by three and two genes respectively [232,233]. Moving onwards, a few research works have brought to the surface the exclusive involvement of only two G-protein subunits in the reception of gustatory stimuli. These monetary elements are Gsα (encoded by DGsα) and Gγ1, both taking part in the tracking of sugars by fruit flies [215]. The reduced gustatory response of flies with faulty Gγ1 to a variety of sugars further hints at its role in the transmission of sweet taste perception [234]. To summarize, although clear evidence for the involvement of G-proteins in the functioning of Grs is inadequate, they might be involved in modulating the ionotropic pathway of taste detection in one way or another (Figure 9A).

### 7.2. Detection of Salty Tastants

The two amiloride-sensitive ENaC channels, i.e., PPK11 and PPK19, are involved in the detection of salts at lower concentrations in larvae, but at lower and higher concentrations in adult *D. melanogaster*. They are expressed in the taste sensing TO of larvae, and in adult flies on the gustatory bristles of the labellum, legs, and wing margins [221]. The role of PPK channels in salt sensing is well-established in larvae [177,222,235,236], but the same has not been shown reliably in case of adult flies. Thus, the PPK channels may not be essential for salt tasting in adult flies [177]. When *D. melanogaster* larvae detect a salty ligand, these ligand-gated, voltage insensitive, depolarizing cation channels lead to a direct influx of Na^+^/K^+^ ions into the GRNs [235,236]. This results in depolarization of their cell membrane and the generation of an action potential. The role of Ir76b has also been reported in low and high salt sensing in adult flies. The Ir76b functions as a constitutively open Na^+^ channel that leads to an intrusion of Na^+^ ions into the cell followed by its membrane depolarization [220,223]. However, a recent report suggests that Ir76b forms a heteromeric complex with the co-receptor Ir25a, to mediate gustatory salt responses [224]. The role of two low-salt cell type receptors—namely Gr64f and Ir94e—with NaCl-specified tuning and two high-salt cell type receptors—i.e., Gr66a and labellar PPK23^glut^—responding to high doses of an array of salty tastants have also come to light [224]. This is how salt signaling works in *D. melanogaster* and helps them respond to salty tastants in an appetitive or aversive manner (Figure 9B).

## 8. Gustation-Based Repugnancy—How Repellents Target the Gustatory System

Apart from triggering an olfaction-based repellency, DEET has also been reported to target the gustatory system of insects with far more sensitivity. It acts as a feeding deterrent. Initially, it was reported that Gr32a, Gr33a, and Gr66a, translated extensively in the bitter sensing GRNs, were involved in the sensing of DEET. However, the expression of these receptors in sugar-detecting GRNs did not trigger DEET repellency. This hinted towards the involvement of some additional players in the manifestation of the aversive behavior [237]. Later on, a work using *D. melanogaster* confirmed the participation of Gr32a and Gr33a in the induction of DEET repugnancy [153]. The gustatory mode of functioning of DEET was also observed when fruit flies devoid of labellum exhibited a decreased avoidance of this repellent. Further in-depth molecular analysis revealed that, in addition to Gr32a, Gr33a, and Gr66a, Gr89a is also involved in DEET repugnancy [238]. To conclude, these findings suggest that the functioning of DEET involves the activation of bitter-sensing GRNs. Nevertheless, a complete repertoire still needs to be accomplished, which will be beneficial in designing improved and more potent alternatives to combat the insect menace.

As mentioned earlier, despite DEET being a potent repellent, its application is associated with the problems of higher costs, unpleasant odor, and reduced efficacy upon recurrent use. Therefore, other alternatives to DEET, such as coconut oil, picaridin, lemon eucalyptus oil, permethrin, etc. are being explored by researchers to develop new repellent compounds with higher efficacy and pleasant smells [23,239]. In this quest, saponins have emerged as compounds that can be used as a substitute for DEET. They are used by plants as a natural means to ward off insects due to their toxicity and unpleasant taste. A saponin extracted from the plant *Quillaja saponaria* has been found to exhibit insecticidal and anti-feeding properties on *D. melanogaster* in the larval stage, as well as the adult stage. Further, through behavioral screening and tissue-explicit rescue trials, the role of Gr28b expressed in the labellum of fruit flies has come to light with no involvement of Gr32a, Gr32b, and Gr66a. More precisely, the Gr28b.c isoform triggers the aversive saponin response [240] (Figure 10). Altogether, it is a great sign, and more such works are needed in this direction to develop better alternatives to DEET by targeting the gustatory system of insects.

## 9. Conclusions & Future Directions

The studies involving olfactory and gustatory processing in *D. melanogaster* have made rapid progress in the last few years. This has further strengthened the position of fruit flies in the arena of neuroscience as a versatile model organism. As a result, they are being used quite prominently to test and develop new olfaction and gustation-based repellent compounds as alternatives to DEET. When we talk about olfaction, we can say that a number of its aspects have been analyzed in detail by the researchers. The major olfactory organs—the OSNs, their receptors, and ligands—have been covered in detail. Still, several gaps need to be filled, both at physiological and molecular levels. The functional characteristics of lateral horns (LHs), the role of Irs in olfaction, their ligands along with the dose-response curves, and the complete molecular circuitry of olfaction need to be explored. As far as gustation is concerned, the research work using fruit flies has shed some light on their feeding and sexual behavior, along with the gustatory circuits involved. More studies are required in this direction to obtain a clear picture of the molecular and signaling pathways leading to the gustatory response in *D. melanogaster*. Moreover, the effect of hunger and satiety on the feeding behavior of flies, their ability to differentiate between the choices of food quality, and the correlation between olfactory and gustatory pathways need to be established more strongly. The information gained will enable researchers to check a much-enlarged pool of natural and synthetic compounds that can be developed into potent repellents and pesticides. It will be a great service to the modern world, which is crippled with the problems of vector-borne diseases and crop damage on a large scale.

## Figures and Tables

**Figure 1 insects-13-00142-f001:**
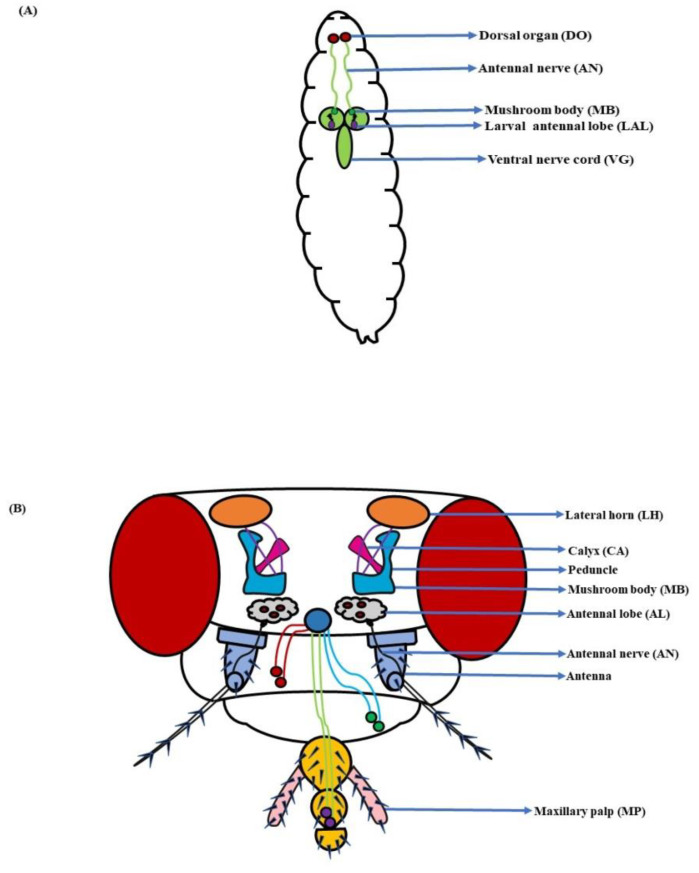
Components of the olfactory system: (**A**) Larva: With a much smaller and simplified architecture, the dorsal organ (DO), antennal nerve (AN), mushroom body (MB), larval antennal lobe (LAL), and ventral nerve cord (VG) constitute the larval olfactory system; (**B**) Adult: The antenna and maxillary palp (MP) along with the antennal lobe (AL), antennal nerve (AN), mushroom body (MB), and lateral horn (LH), complete the fly’s olfactory system (Modified from [36,37,38,39]).

**Figure 2 insects-13-00142-f002:**
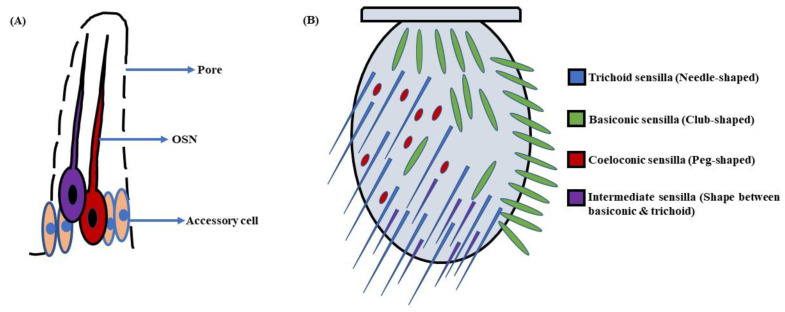
Schematic of (**A**) Sensillum: A sensillum consists of accessory cells, olfactory sensory neurons (OSNs), and pores for odorant molecules to enter; (**B**) Antenna: The antennae in *D. melanogaster’s* head are covered with sensory hairs called sensilla. These are of four types, namely trichoid (needle-shaped), basiconic (club-shaped), coeloconic (peg-shaped), and intermediate (shape between basiconic & trichoid) sensilla (Modified from [40]).

**Figure 3 insects-13-00142-f003:**
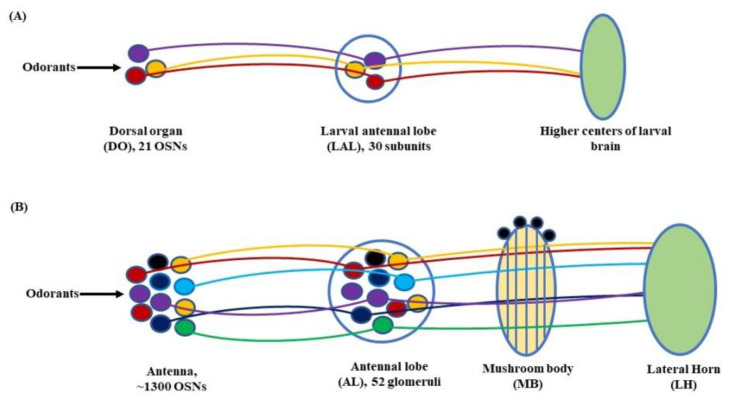
Olfactory signaling pathway of (**A**) Larva: At the larval stage, the odorant molecules are sensed by the dorsal organ (DO). The stimulus is then transferred to the larval antennal lobe (LAL) by the olfactory sensory neurons (OSNs) housed in the DO. From here onwards, through a grid of projection neurons (PNs) and local neurons (LNs) the electrical signal is projected to the larval brain to elicit a response; (**B**) Adult: In fruit flies, the olfactory signaling begins with the antennae and maxillary palps. The OSNs, with the help of odorant receptors (Ors), carry the olfactory cue to the antennal lobe (AL). From there on, through a network of PNs and LNs, the stimulus is transferred to the mushroom body (MB) and lateral horn (LH) to initiate a response (Modified from [34,50]).

**Figure 4 insects-13-00142-f004:**
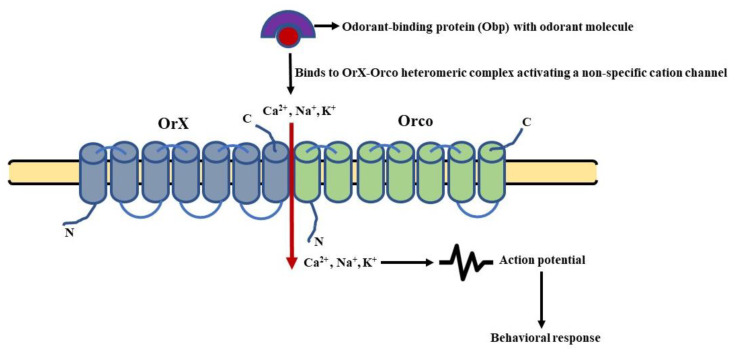
Schematic of ionotropic signaling in general odorant sensing: It involves opening a non-specific cation channel, owing to the binding of odorant molecules to the OrX-Orco heteromeric complex. This leads to a change in the membrane potential and depolarization of the olfactory sensory neurons (OSNs). (Modified from [130]).

**Figure 5 insects-13-00142-f005:**
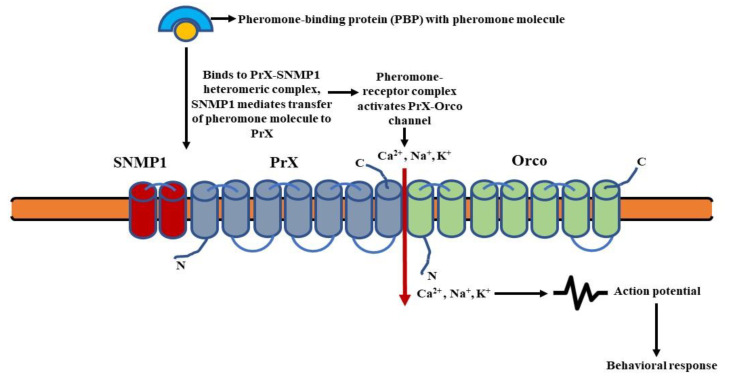
Schematic of ionotropic signaling in pheromone-sensing Ors (Prs): Binding of pheromone molecules to the specific PrX-SNMP1 heteromeric complex results in the activation of an ionotropic channel. This leads to an influx of cations and subsequent depolarization of the pheromone sensory neurons (PSNs). (Modified from [130]).

**Figure 6 insects-13-00142-f006:**
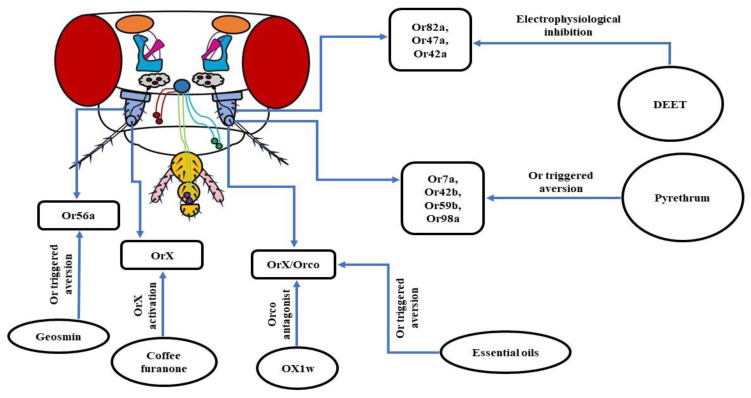
Schematic of molecular players involved in the olfaction-based avoidance of insect repellents: Using *D. melanogaster* as a model organism it has been observed that the functioning of DEET, pyrethrum, and a slew of new potent alternatives is based on the targeting of either OrX or Orco subunit of the OrX/Orco heteromeric complexes.

**Figure 7 insects-13-00142-f007:**
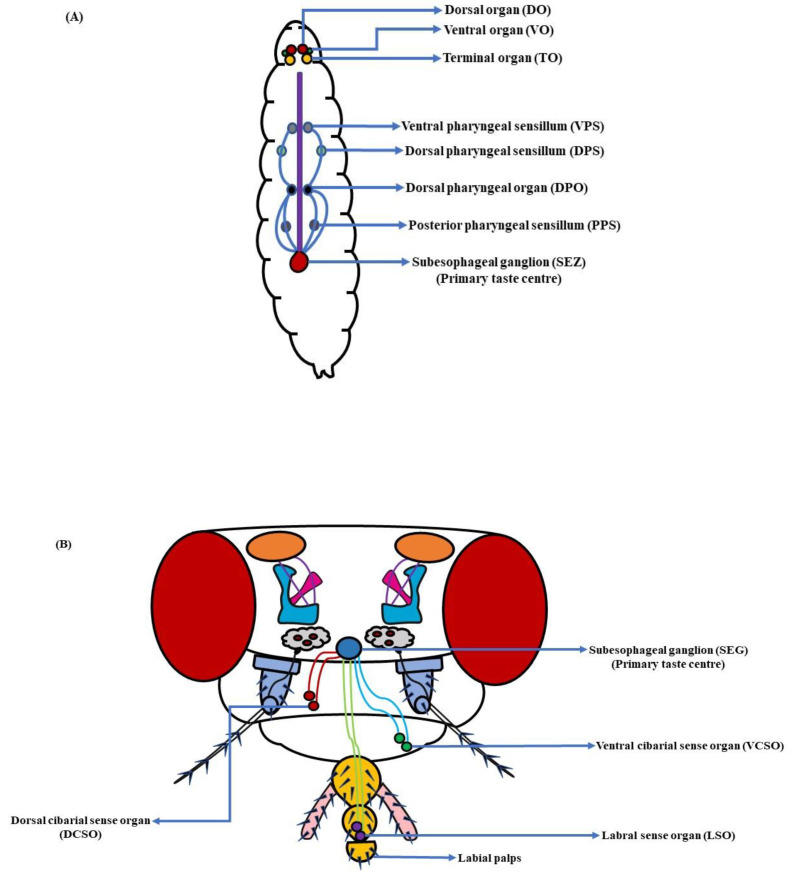
Components of the gustatory system: (**A**) Larva: The taste system in larvae is mainly limited to the head region. The dorsal (DO, peripheral region) terminal (TO) and ventral (VO) organs, along with the dorsal (DPS), ventral (VPS), posterior (PPS) pharyngeal sensilla, and the dorsal pharyngeal organ (DPO) constitute the larval gustatory system. These are all present in sets of two; (**B**) Adult: The mouth part of *D. melanogaster* is composed of the extrinsic and intrinsic taste organs. The labial palps and proboscis are the extrinsic taste organs, whereas the intrinsic taste organs include the labral sense organ (LSO), ventral cibarial sense organ (VCSO), and dorsal cibarial sense organ (DCSO). (Modified from [90,172]).

**Figure 8 insects-13-00142-f008:**
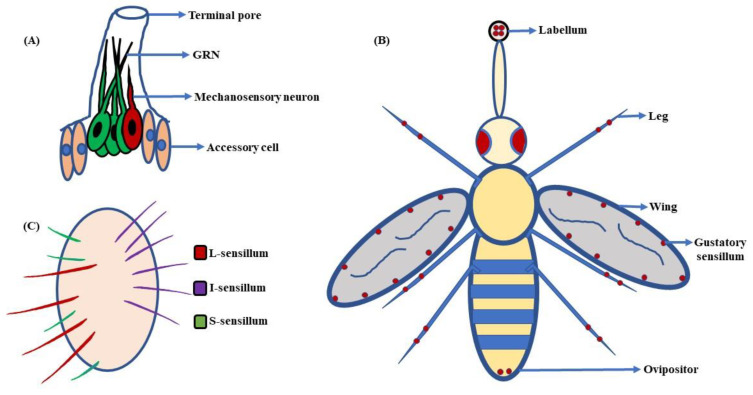
Schematic of (**A**) Gustatory sensillum: A gustatory sensillum consists of accessory cells, gustatory receptor neurons (GRNs), mechanosensory neuron, and a terminal pore for the entry of tastants molecules; (**B**) Distribution of gustatory sensilla on *D. melanogaster’s* body: The gustatory sensilla in fruit flies are scattered on the proboscis, ovipositor, wing margins, and legs; (**C**) Types of gustatory sensilla: The gustatory sensilla, based on their length, are of three types, namely long-type (L-type), intermediate-type (I-type), and small-type (S-type). (Modified from [70,172]).

**Figure 9 insects-13-00142-f009:**
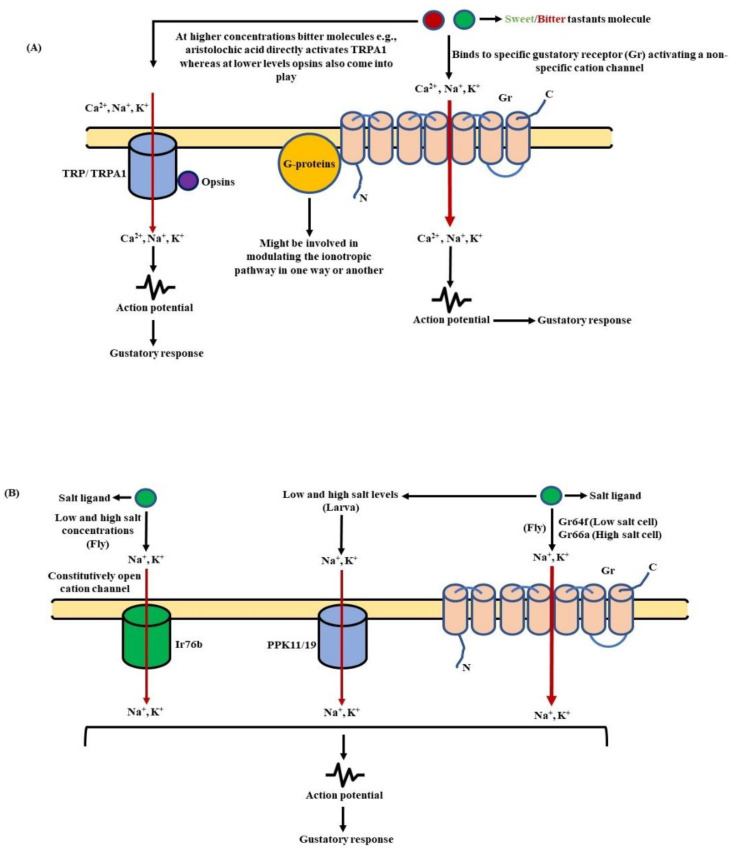
Schematic of taste signaling: (**A**) Sweet/bitter taste signaling pathway: Formation of the receptor–ligand complex activates a non-specific cation channel. This leads to an influx of cations into the GRNs, leading to their membrane depolarization and further transmission of the signal. In addition, bitter tastants such as aristolochic acid utilize TRPA1 and opsins to perform a repulsive gustatory response [202]; (**B**) Salt taste signaling pathway: When *D. melanogaster* larvae detect a salty ligand, the PPK11 and PPK19 channels open up. This leads to an influx of cations into the GRNs, leading to their membrane depolarization. Contrarily, in case of adult flies, Gr64f, Gr66a, and Ir76b are involved in salt signaling (Modified from [181]).

**Figure 10 insects-13-00142-f010:**
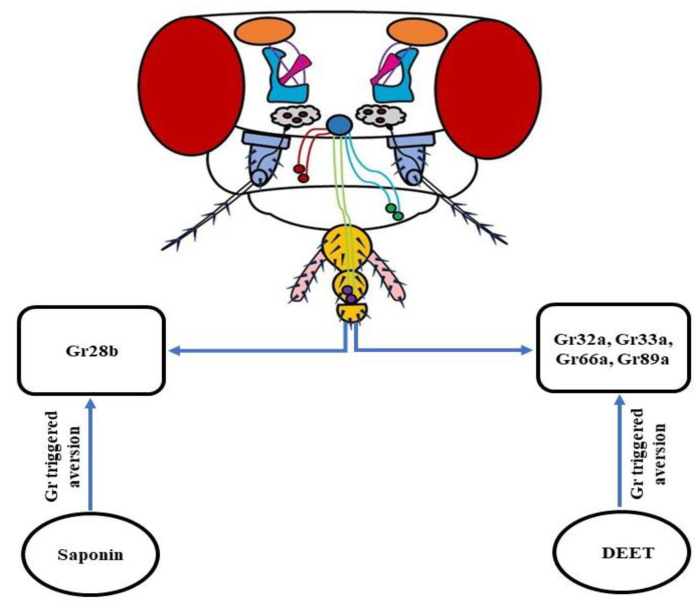
Schematic of insect repellents targeting the gustatory system: Apart from inducing an olfaction-based repugnancy, DEET also functions as a feeding deterrent. Using *D. melanogaster*, it has been observed that DEET and saponins, a class of obnoxious compounds, work by triggering the bitter taste Grs.

**Table 1 insects-13-00142-t001:** Odorant receptors (Ors) in *Drosophila melanogaster*.

Olfactory Sensory Neuron	Odorant Receptor	Glomerulus
-	Or1a *	-
ai3	Or2a	DA4m
ab4A	Or7a	DL5
ab8B	Or9a	VM3
ab1D	Or10a	DL1
ab6A	Or13a	DC2
ai3A	Or19a	DC1
ai3A	Or19b	DC1
ab3A	Or22a	DM2
ab3A	Or22b	DM2
-	Or22c *	-
ai2B	Or23a	DA3
-	Or24a *	-
-	Or30a *	-
ab4B	Or33a	DA2
ab2B, ab5B	Or33b	DM5, DM3
pb2A	Or33c	VC1
ac3B	Or35a	VC3
pb1A	Or42a	VM7d
ab1A	Or42b	DM1
ai3	Or43a	DA4l
ab8A	Or43b	VM2
-	Or45a *	-
-	Or45b *	-
pb2B	Or46a	VA71
ab5B	Or47a	DM3
at4A	Or47b	VA1v
ab10B	Or49a	DL4
ab6B	Or49b	VA5
ab4B	Or56a	DA2
-	Or59a *	-
ab2A	Or59b	DM4
pb3A	Or59c	VM7v
-	Or63a *	-
at4B	Or65a	DL3
at4B	Or65b	DL3
at4B	Or65c	DL3
ab10A	Or67a	DM6
ab9	Or67b	VA3
ab7B	Or67c	VC4
at1A	Or67d	DA1
ab9	Or69a	D
pb1B	Or71a	VC2
-	Or74a *	-
ab5A	Or82a	VA6
-	Or83a *	-
ab, ai, at, pb, ac3	Or83b/Orco	DA, DC, DL, DM, VA, VC, VM
ai2A	Or83c	DC3
ab2B	Or85a	DM5
ab3B	Or85b	VM5d
-	Or85c *	-
pb3B	Or85d	VA4
pb2A	Or85e	VC1
ab10B	Or85f	DL4
at4C	Or88a	VA1d
ab1B	Or92a	VA2
-	Or94a *	-
-	Or94b *	-
ab7A	Or98a	VM5v
ab6	Or98b	VM5d

Outline of odorant receptors (Ors), their OSN classes, and targeted glomeruli. Larval specific Ors are marked with asterisk *. (Adapted from [76,77]).

**Table 2 insects-13-00142-t002:** Ionotropic receptors (Irs) in *Drosophila melanogaster* (larva).

DO	TO	VO	DPO/DPS	VPS	PPS	Abdomen
**Ir21a**	Ir7a	Ir7g	Ir7a	Ir7b	Ir25a	Ir7d
Ir25a	Ir7b	Ir25a	Ir7f	Ir7g	Ir76b	Ir7g
Ir68a	Ir7d	Ir67a	Ir7g	Ir25a	Ir92a	Ir10a
Ir92a	Ir7e	Ir76b	Ir11a	Ir76b	Ir94g	Ir25a
Ir93a	Ir7g		Ir25a		Ir100a	Ir68b
	Ir25a		Ir48b			Ir76b
	Ir56c		Ir48c			Ir85a
	Ir60c		Ir51b			
	Ir76b		Ir60b			
	Ir94e		Ir60d			
	Ir94h		Ir60e			
			Ir67b			
			Ir67c			
			Ir76b			
			Ir92a			
			Ir94a			
			Ir94b			
			Ir94g			
			Ir100a			

Outline of larval repertoire of ionotropic receptors (Irs). (Adapted from [91]). DO—Dorsal organ, TO—Terminal organ, VO—Ventral organ, DPO—Dorsal pharyngeal organ, DPS—Dorsal pharyngeal sensilla, VPS—Ventral pharyngeal sensilla, PPS—Posterior pharyngeal sensilla.

**Table 4 insects-13-00142-t004:** Gustatory receptors (Grs) in *Drosophila melanogaster* (Larva).

DO	TO	DPS	VPS	PPS
Gr2a	Gr2a	Gr2a	Gr28a	Gr22b
Gr28a	Gr9a	Gr22b	Gr33a	Gr28a
	Gr21a	Gr22d	Gr66a	Gr32a
	Gr22a	Gr22e	Gr68a	Gr33a
	Gr22e	Gr23a		Gr39a.a
	Gr28a	Gr28b.a		Gr39a.d
	Gr28b.a	Gr32a		Gr39b
	Gr28b.e	Gr33a		Gr66a
	Gr32a	Gr39a.a		Gr93c
	Gr33a	Gr39a.b		Gr93d
	Gr36b	Gr39b		
	Gr36c	Gr43a		
	Gr39a.a	Gr57a		
	Gr39a.b	Gr58b		
	Gr47b	Gr59d		
	Gr57a	Gr66a		
	Gr58b	Gr77a		
	Gr59a	Gr93a		
	Gr59c	Gr93b		
	Gr59d			
	Gr59e			
	Gr59f			
	Gr63a			
	Gr66a			
	Gr93b			
	Gr94a			
	Gr97a			

Outline of larval repertoire of gustatory receptors (Grs). (Adapted from [154].) DO—Dorsal organ, TO—Terminal organ, DPS—Dorsal pharyngeal sensilla, VPS—Ventral pharyngeal sensilla, PPS—Posterior pharyngeal sensilla.

## Data Availability

The study did not report any such data.

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
