# Peer review of "Drosophila melanogaster Chemosensory Pathways as Potential Targets to Curb the Insect Menace"

_insects, 2022, doi:10.3390/insects13020142_

Round 1

Reviewer 1 Report

This is a rather extensive review of smell in taste receptor in Drosophila. However, there are a number of factual errors that need to be corrected, and a few key points that need to be added. The writing also needs to be improved.

Pg 11-12: Although there is evidence that Gs and Gq contribute to certain aspects of olfaction, the concept that OR activity is coupled to Gs or Gq is quite speculative. The narrative and Figures 4 and 5 indicate that this type of signaling is directly coupled to ORs is widely accepted in the field. This is not the case.

Pg 13: CO2 sensing through coupling of GRs to Gq is also highly speculative and may not be correct. Again, this should not be presented as accepted in the field, and the text and Figure 6a indicate that this is an established view.

Pg 15: There is a full-page discussion of DEET repellency without any mention of the several conflicting models involved in DEET olfactory repellency: masking, confusant, direct activation of receptors. These models should be presented.

Pg 19: The new nomenclature for the different types of GRNs (A-E) should be used here (see Drosophila sensory receptorsa set of molecular Swiss Army Knives. Genetics 2021. 217, 1-34).

L569: It is odd to refer to the detection of “odorants” by the taste system. This should be corrected.

Fig. 9a: Either two or four GRNs should be shown as no sensilla includes three GRNs. They should also be the same color. Add the mechanosensory neuron.

Pg 21: The primary taste center in the brain is called the SEZ, not SEG.

Pg 21: Clarify that GRs are NOT GPCRs.

Pg 23: The very short discussion on sour taste receptors should include two recently published papers on the role of an Otopetrin channel in sour taste. However, there are some major difference between the papers (Molecular and cellular basis of acid taste sensation in Drosophila. 2021. Nat Commun 12, 3730 and Requirement for an Otopetrin-Like protein for acid taste in Drosophila. 2021. Proc. Natl. Acad. Sci. U.S.A. 118: e2110641118.

Pg 22: The description of the role of TRPA1 in sensing aristolochic acid should include the concept that high levels of aristolochic acid are sensed directly by TRPA1 while low levels are sensed through a signaling cascade initiated by opsins and which couples to TRPA1 (Function of opsins in Drosophila taste. Curr. Biol. 2021. 30, 1367-1379.).

Pg 25-26 and Figure 10A: GRs are NOT GPCRs. Rather, GRs are ionotropic receptors. As with ORs, they have an opposite topology to GPCRs. Opsins are the receptors that coupled to the TRPA1 channel in GRNs. The model in Figure 10A is not correct.

Pg 26 and Figure 10B: The roles of Ppk channels in adult salt taste has not been repeated. IR76b is involved in both low and high salt defection. Figure 10B should be corrected.

Pg 26: DPRs affect pathfinding. Therefore, the role of DPR in sensing salt is most likely due to a problem in pathfinding rather than salt reception.

Pg 27-28: It should be mentioned that the gustatory detection of DEET is far more sensitive than olfactory detection of DEET.

Pg 28: The risks of DEET are exaggerated here.

The use of “the” and “a” is incorrect in many places in the manuscript.

L43-44: Unclear what it meant by “have several neurons that we can detect in every individual fly.” There are many neurons that can be detected

L59-60: The following is an incomplete thought. Explain that the pairing is for learning and memory experiments: “Besides, the flies used to effectively avoid odors that were paired with electric shocks [7].”

L70-73: Mention GRs and GRNs are also located in the leg tarsi and around the female ovipositor. Also mention the ovipositor on page 17.

L91-92: Dengue should be mentioned.

L157: There are multiple awkward phrases that should be deleted or reworded. Two out of many examples are “Apart from this” (L157) and “Now coming to” (L164-5).

L167-171: A figure would be helpful to go along with the description of the antennal lobe and wiring to and from the antennal lobe.

Table 1: The no of genes column should be deleted since nearly everything is 1. Also, the co-receptor column should be deleted since ORCO is the co-receptor for nearly all ORs. The few exceptions can be mentioned in the text.

Reviewer 2 Report

The authors have presented a comprehensive review of chemosensory systems in Drosophila, particularly focusing in on two main sections around olfaction and gustation with more detailed subsections beneath each of these. For both chemosensory systems, the authors describe the distribution of sensilla, the main components within the system, and the signalling pathways utilised to direct behaviour. Finally there are sections addressing these chemosensory systems and how they have been studied in relation to pest repellency though olfactory means.

The review is well written, however could do with a polish with minor language edits. For example, on line 74 the authors state “... an augmented echelon of Ca2+ ions...”, which if I have understood this correctly is meaning there is a ‘high level of Ca2+ ions’ present? I suggest simplifying the language here to simply say “high level”.

Another couple of examples of language edits:

Line 113 – “clubbed” change to “separated or divided”

Line 120 – “...stretched but are leaner” change to “...in length but are thinner”

Line 232 – “...IRs also translate into the wings...” change to “...IRs are also found expressed in the wings...”

Line 915 – “...pitch some light...” change to “... shed some light...”

The tables are useful and help to present the large repertoires of ORs, IRs and GRs and their distribution in tissues in an easy to follow manner.

In table 1, I wonder if there is a need to have the OR83b/Orco line in the table? Orco is of course in all sensilla associated with expression of ORs, which you state in the body text, but I think it is a bit confusing in this table. Also the table legend needs to be adjusted to fit the correct style for a legend.

Similarly the figures are generally useful and present the complex signalling pathways in a easy to follow graphical manner. I am unsure if figure 3 is necessary as it does not really add much to the understanding of the two different signaling pathways between larvae and adults, which is explained well in the text.

Heading 4.1 Signaling in food-odor sensing ORs. I suggest this be changed to “general odor sensing” as this would then also include such behaviours as finding oviposition sites which I believe would use the same pathway.

Section 4.2, line 361 – here the authors state that PBP fixes to the PRX-SNMP1 heteromeric complex. I think this should be reworded as the heteromeric complex is really the PR and Orco, the stoichiometry of SNMP1 associating with the olfactory complex is to date unknown. Furthermore, the SNMP1 is likely to only associate with the complex in order to transfer the pheromone from PBP to the OR/Orco ion channel complex.

Reviewer 3 Report

This review presents a comprehensive view of olfaction and gustation in the peripheral neurons in Drosophila. It cites many important studies in the field, although there is an over-reliance on reviews compared to primary research articles.

1. I am very concerned with the inaccurate presentation of GRs and ORs as GPCRs (Section 4.1. and section 7). Although this was the view in the early 2000s (and referenced papers that the authors select), the majority view for the past 10 years is that they are ion channels. Indeed, their inverted topology and lack of any homology to known GPCRs classes makes it impossible to imagine how they could interact with G-proteins nowadays. Most likely they are modulated by G-protein signaling. It is misleading to the field to suggest otherwise. References such as 111 are based on computational modeling but do not provide direct evidence that they are GPCRs. Showing association with G-proteins in multiple figures is a problem. In Section 7, references include a review from 2009 and a paper from 2002 and one in 2003, when it was unknown that these are ion channels. Ref #203 shows that G-protein mutants can modulate olfactory neuron responses, but does directly show that the receptors are GPCRs. Likewise #204 shows expression of G-proteins in the neurons, but not that the receptors themselves are GPCRs. Nearly all cells express G-proteins, but you cannot infer that a receptor is a GPCR by this alone. These ideas must be entirely removed for this review article to be acceptable for publishing. A consideration of their modulation by G-proteins is acceptable if this is desired.

2. The text is repetitive in places and could use tighter organization so that it is more readable. In part this is because first the receptor families are described (such as IRs involved in salt sensation, those in acid sensing, those in sugar sensing, etc), then salt sensation is described There are some typos and grammatical errors that should be corrected.

3. Table 1: at2 and at3 are now being called ai2 and ai3 by most researchers. Also Ir76b is co-expressed, but does not interact with Or35a. See Vulpe Menuz Frontiers 2021.

4. Proper gene names should be used throughout the text. According to the definitive FlyBase, ORs are name Or35a, not OR35a for example. Please use FlyBase names for ORs, Grs, IRs, and Obps in the text.

5. Figure 3 is misleading. PNs project to MB and LH independently not in a series. Likewise, why label 60 ORs when fewer are actually in the antenna. Why not include IRs as well?

6. Section 5 The mechanism of DEET is still debated. Please see review by DeGennaro 2015 for improved understanding of possible mechanisms. Further the paper describing the Ir40a mechanism Ref #136 was retracted and should not be included.

5. Some important references were missed:

  1. Section 2- Olfactory system Components and basic Organization: EM work by Nava-Gonzales and Su (eLife 2021) should be incorporated when describing the sensilla.
  2. Section 2- Olfactory system Components and basic Organization: Some coeloconic sensilla have four ORNs (Vulpe Menuz Curr Biol 2021)
  3. Section 2.1 There is high profile recent work from the Jefferis lab that took a comprehensive view of this topic such as Schlegel Jefferis 2021 and others. Further, Grabe Sachse 2016 counted numbers of PNs and ORNs in glomeruli. The current reference only examined a single gloemerulus. The number of AL glomeruli is thus higher than previously thought.
  4. Section 2.2 When describing Orco and Ors it would be remiss to not fully discuss the cyroEM work from the Ruta lab (2018 and 2021) showing their crystal structures as ion channels, binding pockets, etc.
  5. Section 2.3 Include expression patterns from the comprehensive study of Koh Carlson Neuron 2014.
  6. Section 2.3 The functional studies on IR by Abuin Benton 2011 and 2019 should be incorporated. Also note that some of the 13 IRs are NOT olfactory, but instead hygrosensory or thermosensory.
  7. Line 240 Vulpe Menuz Frontier 2021 should be cited for Ir76b as co-receptor in addition to Abuin Benton 2011
  8. Section 6. You should also describe the mechanosensory neurons in the taste sensilla to be accurate. Such as Sanchez-Alcaniz Benton 2017, Jeong Moon Nat Com 2016
  9. Section 6, page 20. A reference is needed when describing the different leg sensilla. Maybe Ling Carlson #164?
  10. Section 6, p22. Wen discussing bitter taste receptors be sure to include Delventhal Carlson eLife 2016, Dweck Carlson Cur Biol 2020, and perhaps Delventhal Carlson Sci Rep 2017
  11. Section 6.d New work from two labs indicates that Otopetrins are involved primarily Ganguly Montell 2021 and Mi Zhang Nat Comm 2021
  12. Table 5. Interestingly many Grs are in the antenna, not just those listed. Carlson Neuron 2014, Fujii Amrein 2015
  13. Section 6.2. Be sure to include Sanchez-Alcaniz Benton 2018, Koh Carlson 2014, Stewart Carlson PNAS 2015 for the expression patterns
  14. Section 7.2 This should be rewritten. It is now thought that Ir76b and ir25a are co-receptors for the salt receptor (ex Ref 197).

Round 2

Reviewer 1 Report

The review is improved. The main remaining issue is that the review needs to be carefully edited to eliminate the many grammatical errors and awkward word choices. A few minor issues are listed below.

Line 73: Change “a high level” to “an increase.”

OR and IR rather Or and Ir when referring to olfactory receptor and ionotropic receptor proteins. Along the same line ORCO rather than Orco protein. The rule is that all letters for othe protein abbreviation are uppercase when the full name includes multiple words.

Line 904: Add the reference: Leung et al Function of opsins in Drosophila taste. Curr. Biol. 2021. 30, 1367-1379.

Reviewer 3 Report

The authors have improved the article through minor revisions and inclusion of some additional references. Substantial problems still remain.

1) Most importantly, the article still sounds confused when regarding the role of Ors/Grs as G-proteins instead of their known role as ion channels (for example line 860 "might be functionining as ionotropic receptors". It needs to be clearer that these are definitely ion channels, and that G-proteins likely modulate their signaling (similar to what happens with many ion channels), rather then that they activate G-protein alpha subunits to initiate a signaling cascade. In addition to the text, several figures are misleading in this regard. This is still my primary concern and the reason for requesting a major revision.

2. The English and sentence phrasing is often awkward, and typos and grammatical errors abound. Proofread carefully or consult with an editing service.

3. References to at2/at3 were removed and replaced with ai2/3. However, there should be further consideration of removing ai1 from the related text as there is little evidence that such sensilla exist. The Potter/Lin paper merely suggested that Or13a is in a different sensilla, but did not do structural studies to confirm that it has a single neuron or that it is an intermediate sensilla. Structural studies have not found evidence for an ai1

References- some still need improvement

  1. Section 2- Olfactory system Components and basic Organization: EM work by Nava-Gonzales and Su (eLife 2021) should be incorporated when describing the sensilla. This was cited, but their findings were not incorporated conceptually into the text.
  2. Section 2- Olfactory system Components and basic Organization: Some coeloconic sensilla have four ORNs (Vulpe Menuz Curr Biol 2021). The information was added, but the wrong reference was cited (should not be Vulpe Menuz Frontiers 2021).
  3. Section 2.1 There is high profile recent work from the Jefferis lab that took a comprehensive view of this topic such as Schlegel Jefferis 2021 and others. Further, Grabe Sachse 2016 counted numbers of PNs and ORNs in glomeruli. The current reference only examined a single glomerulus. The number of AL glomeruli is thus higher than previously thought. This was not updated and still references a single study on a single glomerulus.
  4. Section 2.2 When describing Orco and Ors it would be remiss to not fully discuss the cyroEM work from the Ruta lab (2018 and 2021) showing their crystal structures as ion channels, binding pockets, etc. Still not included.
  5. Section 2.3 Include expression patterns from the comprehensive study of Koh Carlson Neuron 2014. Needs to be added to Table 3 and associated text.
  6. Line 240 Vulpe Menuz Frontier 2021 should be cited for Ir76b as co-receptor in addition to Abuin Benton 2011. Shoudl be added to line 329 since Ir76b is a third co-receptor along with Ir25a and Ir8a
  7. Table 5. Interestingly many Grs are in the antenna, not just those listed. Menuz Carlson 2014, Fujii Amrein 2015. The information was added to the table but no references to support this were added
